# The never-ending patient journey of chronically ill patients: A qualitative case study on touchpoints in relation to patient-centered care

Vera K. Maas[1], Frederik H. Dibbets[2], Vincent J. T. Peters[1,2]*, Bert R. Meijboom[2,3], Daniëlle van Bijnen[1]

1 Department of Internal Medicine, Catharina Ziekenhuis, Eindhoven, Noord-Brabant, the Netherlands,
2 Department of Management, Tilburg School of Economics and Management, Tilburg University, Tilburg, Noord-Brabant, the Netherlands, 3 Department of Tranzo, Tilburg School of Social and Behavioral Sciences, Tilburg University, Tilburg, Noord-Brabant, the Netherlands

* v.j.t.peters@tilburguniversity.edu

## Abstract

### Background

Healthcare professionals caring for chronically ill patients increasingly want to provide patient-centered care (PCC). By understanding each individual patient journey, they can significantly improve the quality of PCC. A patient journey consists of patient interactions, so-called touchpoints, with healthcare professionals distributed over three periods: pre-service, service, and post-service period. The aim of this study was to ascertain chronically ill patients' needs for digital alternatives for touchpoints. Specifically, we aimed to explore which digital alternatives patients would like to see implemented into their patient journey to help healthcare professionals providing PCC.

### Methods

Eight semi-structured interviews were conducted either face-to-face or via Zoom. Participants were included if they had visited the department of internal medicine and had received treatment for either arteriosclerosis, diabetes, HIV, or kidney failure. The interviews were analyzed utilizing a thematic analysis approach.

### Results

The results suggest that the patient journey of chronically ill patients is a continuous cycle. Furthermore, the results showed that chronically ill patients would like to see digital alternatives for touchpoints implemented into their patient journey. These digital alternatives consisted of video calls, digitally checking in before a physical appointment, digitally self-monitoring one's medical condition and personally uploading monitoring results into the patient portal, and viewing their own medical status in a digital format. Particularly, patients who were familiar with their healthcare professional(s) and were in a stable condition mostly opted for digital alternatives.

**Data Availability Statement:** We have uploaded our minimal underlying dataset to Tilburg University's data repository, TiU Dataverse. This

repository has been certified with the CoreTrustSeal. You can access the repository via https://doi.org/10.34894/P8FWAL.

**Funding:** The author(s) received no specific funding for this work.

**Competing interests:** The authors have declared that no competing interests exist.

## Conclusion

In the cyclical patient journey, digitalization can help put the wishes and needs of the chronically ill patients at the center of care. It is recommended that healthcare professionals implement digital alternatives for touchpoints. Most chronically ill patients consider digital alternatives to lead to more efficient interactions with their healthcare professionals. Furthermore, digital alternatives support patients to be better informed about the progress of their chronic illness.

## Introduction

Healthcare professionals caring for chronically ill patients increasingly want to provide patient-centered care (PCC), given that patients are demanding to have more influence and control on the treatment they receive. PCC has been a focus in healthcare research for a significant amount of time to make care more focused on the patient and thereby improve safety, efficacy, public health, and patient satisfaction [1]. PCC focuses on the patient's healthcare needs and has the goal to empower patients to become active participants in their care [1]. PCC could possibly replace the current physician-centered system with one that revolves around the patient [2]. This is important because each patient is unique and requires treatments that suit their situation in a patient-centered way. This implies that due to continuously changing patient needs and wishes, healthcare professionals are required to innovate and change the way they deliver care regularly. Also from a societal perspective, it is important that healthcare services become more responsive to the complex needs and requirements of chronically ill patients [3]. Additionally, hospitals should keep up with the ever-increasing amount of diagnostic and treatment options, specifically including digital opportunities. Gerteis et al. [4] introduced seven dimensions that affect PCC: 1) Respect for patients' values, preferences, and expressed needs, 2) Coordination of care, 3) Communication between patient and providers, 4) Physical care, 5) Emotional support, 6) Involvement of family and friends, and 7) Transition and continuity of care. These dimensions provide insights in how healthcare professionals can become more involved in patient-centered quality efforts, and how patient-centered quality can be integrated into health care policy, standards, and regulations. By bringing the patient's perspective to the design and delivery of health services, providers can improve their ability to meet patient's needs and enhance the quality of care [4].

Hospitals can also significantly improve the quality of the service provided by exploring and understanding the individual patient journey [5]. The patient journey principle stems from the customer journey principle, an often studied topic within service and marketing research [6]. The customer journey is defined as "the customer's personal interpretation of the service process and their interaction and involvement with it during their journey or flow through a series of touchpoints" [6]. In the context of healthcare, the patient journey represents patient interactions, i.e., touchpoints, with multiple healthcare professionals distributed over space and time [7].

Touchpoints are present in every patient journey and are widely discussed in service research [6]. A touchpoint is considered as a moment of contact between a firm or service provider and a customer at distinct points in the customer journey [8]. Touchpoints differ from each other in the way the contact between the patient and the healthcare professional is established. In this study, the focus will be on two categories of touchpoints. The first category of touchpoints is described as healthcare encounters [9]. These encounters could be hospital

admissions, urgent care, and primary care provider visits. For these types of touchpoints, the patient is usually required to visit the healthcare institution or healthcare professional. Some of these healthcare encounters that are currently taking place physically at the hospital could also be performed by the patient. For example, the self-monitoring of blood glucose for diabetes patients which has been around for years now has positively impacted the self-management of this illness [10].

The second category of touchpoints are described as device touchpoints and are considered technological solutions that are utilized by patients and healthcare professionals [9]. These touchpoints involve, for example, online meetings with healthcare providers, administrative tasks such as planning appointments, and informational actions like reviewing one's medical records. This category of touchpoints provides tremendous opportunities to improve patient experiences and possibly also improve the efficiency of care [11]. For example, the use of video calls was found to have the potential to improve the patient experience especially if these patients otherwise must travel long distances [12].

The patient journey can be divided into three specific periods: pre-service, service, and post-service [13]. A basic patient journey of chronically ill patients usually begins during the pre-service period with the registration of the patient at the outpatient clinic. Following, an intake consult takes place between the patient and the healthcare professional to establish a first relationship and discuss the expectations for the treatment. In numerous hospitals the physical check-in at the desk before an appointment has been replaced by digital registration screens, which is found to be more efficient and less expensive compared to having several staff members at the desk. Furthermore, it improves the general patient satisfaction [14].

In the service period of the patient journey, the focus is on the regular checkup-meetings patients with chronic illnesses tend to have. Typically, chronically ill patients have a low chance of being cured. Therefore, the main meetings they have are with their healthcare professional to check their health status and to discuss the results of previous blood tests. However, extensive physical examination is usually not needed. Due to the Covid-19 pandemic physical meetings with healthcare professionals have increasingly been rescheduled to an online format. Ramaswamy et al. [15] showed that video appointments had been increased by 8729% during the pandemic compared to the year before. They also found that patient satisfaction with video appointments is very high and is not a barrier to shift from physical visits at the outpatient clinic to online video visits. In addition, research shows that most patients have the required knowledge and technology to participate in online checkup meetings [16].

The post-service period of the patient journey is dominated by administrative actions. These involve the planning of new appointments and other administrative tasks, as well as reviewing one's medical records. A study on patient access to medical records has found modest benefits and showed moderate improvements in doctor-patient communication, adherence, patient empowerment, and patient education [17]. Though, it was also found that some patients experienced slight confusion while reading their own records and could not understand everything that was in their file. Currently various applications of digital solutions in hospital care have been integrated into so-called online patient portals. Research has indicated that the use of these portals can lead to improvements in clinical outcomes, patient behavior, and experiences [18].

Previous analysis of the patient journey has revealed three important needs for PCC, especially for patients with chronic conditions [19]. First, the patient and the healthcare professional should develop a visible health goal. Second, they should establish clear and transparent shared decision-making. Third, they should have a closed-loop communication process [19]. More factors, for example the implementation of digital innovations, can be identified in order to create a better setting for PCC [20]. Digital developments can help improve healthcare

service by improving connectivity between the patient and the healthcare professional [21]. Currently, little is known about the perspective of chronically ill patients on the implementation of digital opportunities of their patient journey to improve the delivery of PCC. Although we acknowledge the eHealth Enhanced Chronic Care Model, as proposed by Gee et al. [22], which places chronic care in the context of the community where the person will receive care, we are convinced that the work by Gerteis et al. [4] provides a more comprehensive view on how digitalization can improve dimensions of PCC.

The main aim of the present study is to ascertain the chronically ill patients' needs for digital alternatives for touchpoints between them and their healthcare professionals. Specifically, we aim to explore which digital alternatives the patients would like to see implemented into their patient journey to help healthcare professionals providing PCC. The potential digital alternatives for the touchpoints are allocated to the three periods of the patient journey: pre-service, service, and post-service period. Each period has its own characteristic touchpoints and potential digital alternatives. The digital alternatives that chronically ill patients like to see implemented can lead to higher levels of patient satisfaction as it responds to the patient's own needs and preferences, and thus we assume that these alternatives can improve the delivery of PCC.

## Methods

### Study design

A qualitative, exploratory case study design was used to identify the touchpoints between chronically ill patients and healthcare professionals during the patient journey and to explore opportunities for digital alternatives for these touchpoints to make the patient journey more patient-centered. Because the topic of this study is in its formative stage, we conducted qualitative research in the form of a case study. Case study research designs have different purposes, namely exploratory, explanatory, and descriptive, or a combination of those purposes [23]. The design of the current study can be defined as exploratory, as it explores the patient journey of chronically ill patients, their attitudes towards that journey, and the possible digital alternatives provided by these patients for the encountered touchpoints during their journey. We used a case study approach to generate an in-depth understanding of a complex issue in its real-life context, which lends itself well to capture information on 'how', 'what', and 'why' questions [24]. An advantage of conducting research in this manner is that it provides the opportunity to study a topic in its real-life context; this can contribute to understanding whether digital alternatives for touchpoints make the patient journey of chronically ill patients more patient-centered. The consolidated criteria for reporting qualitative research (COREQ) [25] were used as guidelines for the study design and the data analysis (S1 File).

### Context

We took the healthcare provision for chronically ill patients characterized by continuous, monitoring episodes, provided at the department of internal medicine of a hospital in the south of the Netherlands, as our case. The department consists of outpatient clinics for patients who suffer from (chronic) illnesses such as infectious diseases, kidney diseases, metabolic diseases, and vascular diseases. The case sampling of the participants focused on typical cases of care for chronically ill patients characterized by continuous, monitoring episodes at this department. In the fall of 2021, using e-mail and telephone, the study team approached three internists at the department of internal medicine to help us select and contact potential participants. Potential participants were considered for inclusion if they visited the department of internal medicine and are treated for either arteriosclerosis, diabetes, HIV, or kidney failure.

## Participants

Chronically ill patients who received treatment and are monitored at the department of internal medicine were asked to participate. First, convenience sampling logic was used to identify potential participants [26]. Second, purposive sampling was used to target specific chronic illnesses to capture a wide variety of experiences of patients who are being treated at the department of internal medicine, with a focus on patients who suffered from infectious disease (HIV), kidney disease (kidney failure), metabolic diseases (diabetes), and vascular diseases (arteriosclerosis). Table 1 provides an overview of the demographic profile of ages of these four patient groups. Potential participants were given two weeks to consider whether they wished to participate and, in the case of a positive decision, were asked to reply to the internist and give consent for their contact details to be disclosed to the study team. The participants who gave consent were then contacted by telephone to schedule the interview. The sample size aim was to include at least three patients of each patient group, so that we would conduct twelve interviews in total. This number of interviews was based on previous research that proves that data saturation, meaning new interviews do not yield new data on the interview topics, can be achieved with 9 to 17 interviews [27, 28]. Eight patients agreed to participate in this study, despite our continued efforts to include more patients. Although the number of participants is relatively small, a limited number of interviews can be sufficient in the case of exploratory studies to get a reliable sense of thematic exhaustion and variability of the data [27, 28]. In addition, research has indicated that sample size should not be considered alone but be embedded in the more encompassing examination of data adequacy [29]. This implies that sample size numbers in qualitative research are not unimportant but should be extended to terms of adequate amounts of evidence, adequate variety in kinds of evidence, and adequate interpretive status of evidence. Given our comprehensive data collection, consisting of semi-structured interviews and document analysis, we are convinced that, despite our relatively small number of interviews, we were able to present adequate variety and interpretation of our evidence. By combining the information from the interviews and document analysis with the theoretical framework, we were able to draw a comprehensive image of the patient journey for the patient groups included in our study. For instance, when an interviewee mentioned a specific touchpoint, we verified this touchpoint with our collected documents and interview transcripts to ensure that this touchpoint was in accordance with the baseline patient journey we developed. This approach strengthened us in our belief that we were able to present convincing evidence. This conviction also arose during the data analysis when we observed that data saturation happened after seven interviews, as no new themes emerged from the data gathered between interview seven and interview eight. This is in accordance with previous research that demonstrated that saturation can be achieved in a narrow range of interviews [27].

## Data collection

The study team approached twelve chronically ill patients of the department of internal medicine. Eight of them agreed to participate in the present study, none of them were from the

**Table 1. Demographic profile of ages of patient groups by disease.**

| Number of patients | Patient group by disease | Average age | Standard deviation | Minimum age | Maximum age |
|---|---|---|---|---|---|
| 12 | Arteriosclerosis | 62.01 | 11.65 | 42 | 76 |
| 1535 | Diabetes | 58.43 | 22.29 | 17 | 95 |
| 615 | HIV | 47.75 | 18.45 | 19 | 88 |
| 598 | Kidney failure | 71.62 | 20.84 | 19 | 97 |

**Table 2. Participant characteristics.**

| Participant | Chronic illness | Gender | Age | Number of years living with chronic illness |
|---|---|---|---|---|
| 1 | Diabetes | Female | 42 | 20 |
| 2 | HIV | Male | 53 | 16 |
| 3 | Kidney failure | Female | 78 | 13 |
| 4 | Kidney failure | Female | 51 | 1 |
| 5 | Diabetes | Female | 41 | 38 |
| 6 | Kidney failure | Male | 56 | 29 |
| 7 | HIV | Male | 41 | 12 |
| 8 | HIV | Female | 38 | 15 |

cardiovascular patient group. Reasons for not participating that were given were no interest in participation or not having sufficient time for the interview. Table 2 provides an overview of the participant characteristics. Three of the participants were male and five of them were female. The youngest participant was 38 years old and the oldest was 78 years old ($M$ = 50.00 years; $SD$ = 13.09). On average, participants were living with their chronic illness for 18 years at the time of the interviews ($M$ = 18.00 years; $SD$ = 11.26). The interviews lasted from 20 to 55 minutes.

We conducted eight semi-structured interviews which allowed us to make sure that important topics were covered, while leaving room for the participants to provide their story regarding their patient journey [30]. Given the qualitative and exploratory nature of our case study, the focus was primarily on understanding the experiences, perceptions, and attitudes of chronically ill patients during their patient journey. The semi-structured interview approach is relevant for exploratory research, as it is an effective way to gather rich data and it allows for the creation of new insights into the case under study [23]. As a consequence of Covid-19 restrictions in the Netherlands in 2021, five interviews were conducted via Zoom [31] instead of face-to face. We also collected patient characteristics such as age, gender, type of disease, and how long the participants were living with their chronic illness.

The interview questions were grouped in advance according to the three phases of the patient journey: pre-service period, service period, and post-service period [13]. In each interview the same questions were asked, but not in a fixed order (Table 3). The semi-structured approach was used to obtain insights in the touchpoints in the patient journey during different periods. In addition, we collected participants' needs, preferences, and suggestions for digital alternatives for the touchpoints per period of the patient journey. The interviews were audio recorded and transcribed verbatim.

We also collected relevant documentation that was open to the public (e.g. information leaflets, invitation letters), and internal documentation of the department of internal medicine (e.g. planning schemes, medical protocols). These collected documents provided valuable information in terms of touchpoints encountered by chronically ill patients during their patient journey.

## Data analysis

The data were analyzed using the three steps method for thematic analysis as described by Miles et al. [32]: 1) data reduction, 2) data display, and 3) drawing conclusions. This is a systematic data reduction process building on, among others, the reading of transcripts, document summaries, codification of text segments, generation of themes and categories, and identification of relationships [32]. We employed this deductive, thematic analysis approach

**Table 3. Interview questions.**

| General questions | 1. What is your age? |
|---|---|
| | 2. What is your gender? |
| | 3. How many number of years are you living with your chronic illness? |
| | 4. In general, what do you think about digitization in healthcare? |
| | 5. What are the advantages of digital solutions for you? |
| | 6. What do you see as the disadvantages of digital solutions? |
| | 7. In what areas do you think digital applications would be most beneficial? |
| | 8. To what extent are you already using digital solutions or applications? |
| | 9. What do you think about when it comes to putting the patient first? |
| Patient journey: Pre-service period | 1. You have been referred to the specialist and then have an intake interview. How were you informed beforehand? Was this to your satisfaction? |
| | 2. Were there any surprises for you? Could you have been better informed on this? |
| | 3. In general, are you well prepared for what is to come? How is this done? |
| | 4. Could digital education do something for this? |
| | 5. What would you think of being able to send in questions as part of the preparation? |
| | 6. What possibilities do you see for this yourself? |
| Patient journey: Service period | 1. Was it also clear what to expect? |
| | 2. How often do you have a check-up at the hospital? |
| | 3. Was this also the case in corona time? Did you notice any differences? |
| | 4. Would you be comfortable doing some operations yourself? |
| | 5. How do you usually make an appointment? |
| | 6. Is that the easiest for you, or would digital also be a practical solution? (fill in your own results) |
| | 7. Do you have experience with online appointments and video calling? |
| | 8. Were you prepared for this? |
| | 9. What do you think are the advantages of a digital appointment? |
| | 10. What do you think are the disadvantages of a digital appointment? |
| | 11. What would such a digital solution need to meet for you? |
| | 12. Do you notice that your doctor is positive about this? |
| | 13. How would you and your family feel about being able to participate? |
| | 14. For example, could a recorded video or audio recording of the conversation help you better remember the content of the conversation? |
| | 15. Would it be more convenient for you if there were broader timeslots to have an appointment with the hospital? |
| Patient journey: Post-service period | 1. Is it easy to see your own file? Have you ever had trouble doing this? |
| | 2. Do you look for things on the internet or certain online forums? |
| | 3. Can you easily ask your doctor or nurse questions outside of appointments? |
| | 4. Would this be easier for you if you could do this online? |
| Hospital digitization | 1. When you register yourself at the hospital, do you do so through the front desk? |
| | 2. How would you feel if this were done digitally? |
| Closing | 1. Do you have any ideas of your own about digitization in the hospital? Are there things you miss or would like to see implemented? |
| | 2. What are the barriers to digital opportunities for you? |
| | 3. What are the drivers for digital opportunities for you? |

since we used a coding framework for analysis that was based on concepts and definitions derived from the literature [4, 6, 13, 19]. While exploratory case studies probe into and shed light on what is essentially unknown, they should be guided by a specific purpose that frames the research [23]. The deductive codes were useful in both the segmentation and coding phase

**Table 4. Coding scheme.**

| Main category | Subcategory | Code | Definition | Sub-code |
|---|---|---|---|---|
| Touchpoints | Healthcare encounters | HC | Encounters that require physical presence and examination. | HC-B Blood tests<br>HC-C Checkup meeting<br>HC-O Other healthcare encounters |
| | Device touchpoints | DT | All touchpoints that are related to a digital contact moment with the hospital. This could be to obtain practical information or planning, medical meetings or to review one's medical records. | DT-IP Informational or planning related<br>DT-M Medical contact<br>DT-F Review of one's medical records and results. |
| Dimensions that affect Patient-Centered Care (PCC) | Respect for patients' values, preferences, and expressed needs | RPV | Acknowledging the patient as a person and recognize their unique qualities, personal values, beliefs, boundaries, and perspectives. | |
| | Coordination of care | CC | integration of services within an institutional setting. | |
| | Communication between patient and providers | CPP | Dissemination of accurate, timely and appropriate information; and education about the long-term implications of disease and illness. | |
| | Physical care | PC | Comfort and the alleviation of pain. | |
| | Emotional support | ES | Alleviation of fears and anxiety. | |
| | Involvement of family and friends | IFF | Active involvement of and support for the patient's relatives and friends to the degree that the patients prefer. | |
| | Transition and continuity of care | TCC | Facilitation of healthcare that is well coordinated and allows continuity. | |
| Digital solutions | Medical interactions | MS | All solutions where a healthcare professional must be consulted. | |
| | Self-monitoring | SS | Self-monitoring solutions, for example, uploading results. | |
| | Administrative | AS | Solutions that have to do with non-medical device touchpoints. | |
| Other | Other touchpoints | OT | Touchpoints that cannot be characterized through the descriptions above. | OT-L Letter of the hospital<br>OT-O Touchpoints outside of the medical patient journey. |

of the data analysis. Three main categories were created to store the coded data: touchpoints, patient-centered care, and digital solutions. The definitions of these categories and their sub-categories can be found in Table 4. These (sub)categories were later used to label the data with the relevant codes. The coding framework was continuously discussed and tested during the coding of the interviews, which is in alignment with the main principle that Miles et al. [32] endorse and entails that codes should 'have some conceptual and structural unity. Codes should relate to one another in coherent, study-important ways; they should be part of a uni-fied structure' [32].

We started with developing a baseline patient journey, which is common practice in patient journey research [33], based on the data commonalities that were obtained from the interviews (S1 Fig). Next, as our guiding principle, we used the touchpoint definition in customer jour-neys of Lemon & Verhoef [6] to identify touchpoints in the patient journey: a moment of con-tact between a firm or service provider and a customer at distinct points in the customer journey (p. 71). It was then discussed within the research team whether these touchpoints could be described as healthcare encounters or device touchpoints. To obtain more detailed information about the touchpoints and about the moments at which they occurred in the patient journey, the touchpoints were grouped together according to the three periods of the patient journey, as identified by Rosenbaum et al. [13]. After the coding process, we systemati-cally analyzed the touchpoints to explore whether digital alternatives that were provided by the participants could be used in the patient journey and how this could improve PCC, using the dimensions identified by Gerteis et al. [4].

## Validity and reliability

To establish validity and reliability of our data, several measures were taken [34]. To improve the internal validity, the interviewed patients were initially contacted by their healthcare professionals to ask whether they would take part in this research. The study and its purpose were explained. To further increase reliability, the interviews have been conducted in Dutch, the native language of the participants, so that no language barriers existed. Additionally, the member checking method was applied. This method entails to check that the transcription was correctly done by returning the transcripts of the interview to the participants [35]. The transcripts were returned to all participants and they were given one week to review and return their transcript. A reminder was sent after one week. We received no comments and corrections, all participants agreed on their transcript.

## Ethical considerations

The Ethics Review Board of Catharina Hospital Eindhoven thoroughly evaluated and approved our study design (nWMO-2021.055). We informed the participants about the study and their rights as a participant in scientific research. All participants provided oral and written informed consent.

## Results

The interview results will be presented according to the three periods of a patient journey: pre-service, service, and post-service period. The touchpoints that are experienced during these periods will be illustrated. Additionally, we show whether the participants are interested in digital alternatives and which digital alternatives they would like to see implemented to improve the delivery of PCC.

## Pre-service period

**Finding the way in and around the hospital.** The interviews provided insights into touchpoints that were not directly related to the patient journey within the hospital. It was mentioned by participants that parking around the hospital can be tremendously inconvenient and time-consuming. Furthermore, finding the way around the hospital was perceived to be difficult too, especially when one must move through the hospital in areas that have not been visited before. To reduce the unnecessary stress that participants experienced about finding the way in and around the hospital, participants would have liked to see digital developments implemented.

*"I believe that one of the areas where the hospital can be improved is the logistics. I think it would be great to be able to check in with your car license plate or pay the parking fee online"*

*–Participant 6*

*"Recently, I had to walk a route in the hospital that I didn't know, so I struggled to find the fastest route. It would have been useful to have some navigation through an app for example."*

*–Participant 7*

**Intake consult with the healthcare professional.** In general, participants stated that they appreciated physical consults, especially when they did not yet had a strong relationship with their healthcare professional. In the intake consult, the first contact between the patient and the healthcare professional took place, a treatment plan was made together with the patient,

and their treatment-related expectations were discussed. Participants indicated that they appreciated this approach.

> *"I would like to see someone if I meet them for the first time. I think it truly matters whether you already have a relationship with the doctor."*
>
> *–Participant 8*

The participants mentioned that they usually received several information documents, such as information leaflets, before the intake consult took place. Some participants pointed out that digital alternatives could have helped them in order to be better prepared for the intake consult.

> *"Now you get an entire library of documents to read, which I did not manage to read. I think some informative clips could work better."*
>
> *–Participant 4*

> *"I am a huge fan of short videos. I like to watch them way more than I like to read a folder, which I usually lose soon after I got it."*
>
> *–Participant 5*

A good preparation before the intake consult took place was important in order to ask the healthcare professional better questions related to the treatment for their chronic illness.

> *"It is important to be well-informed before you meet the doctor. Otherwise, it might be unclear where the difficulties are and what things are important to take into consideration."*
>
> *–Participant 4*

In contrast, another participant stated that she did not prepare the intake consult at all. She specified that she trusts the healthcare professional's judgment and did not want to interfere. This participant mentioned that she accepted the treatment proposals without extensive consideration.

> *"I let the doctor decide everything, as he knows best. I would not know how things work."*
>
> *–Participant 3*

**Check-in at the desk.** Most participants indicated that they did not consider the physical check-in at the desk as a value-added activity. However, one participant did mention that this touchpoint was value-adding.

> *"The people at the desk know me well, so I am able to have some small talk with them."*
>
> *–Participant 3*

All other participants saw no or little added value of the physical check-in at the desk and addressed their interest for a digital alternative for this touchpoint.

> *"I think checking in at the desk is not necessary, digitally would be preferable for me, for example through an app on my telephone."*
>
> *–Participant 6*

## Service period

**Blood tests and checkup meetings in the hospital.** In this period of the patient journey, most of the touchpoints can be considered healthcare encounters. Each chronically ill patient was required to undergo periodical blood tests, which were followed up by the healthcare professionals in a checkup meeting. The regularity of these tests varied from every two months to every six months. The appointments for these blood tests could already be made digitally by the patients themselves and could usually still be scheduled on the same day. However, it was possible to have blood tested without an appointment from 8 a.m. to 7.30 p.m. on weekdays. Participants mentioned that it was reported by the physician's assistants that the waiting line at the blood test center could go up to over an hour when no digital appointment was made.

*"I have to say that the people in the hospital really said that you should make an appointment, otherwise you could be waiting for over an hour."*

*–Participant 7*

*"At this point, you can make a digital appointment yourself, which I think is really convenient. I can book an appointment 45 minutes from now, it is really easy to plan it yourself."*

*–Participant 8*

One participant mentioned that when her blood sample was taken, she had to wait for up to 1.5 hours before the results were in, which were then discussed with the healthcare professional during the physical checkup meeting. The participant indicated that she preferred a digital alternative.

*"Most of the time I have to wait for 1 to 1.5 hours for my blood results. If possible, I would prefer to go home and get the results digitally."*

*–Participant 1*

Another participant stated that he usually had to wait for more than two weeks to receive all the blood results. When he received these results, he usually had to come back to the hospital once again to discuss them physically with the healthcare professional.

Due to Covid-19 restrictions in the Netherlands almost all checkup meetings were conducted via telephone only, to limit the number of people that were visiting the hospital. This was perceived as more convenient by most of the participants because they did not had to wait in the hospital for the results.

*"During Covid-19 I had my blood tested at the hospital. But then they just called me and asked me how I was doing. I liked it, that they called me."*

*Participant 3*

The digital alternative for the checkup meetings via telephone was also discussed during the interviews. Several participants indicated that they would have liked to have a video call with their healthcare professional.

*"Especially if you know most of the routine appointments, it would be nice for me if things like that could be done online."*

*–Participant 5*

One participant indicated that she had two separate checkup meetings in the hospital, one appointment with her doctor and one appointment with the nurse. When the possibility of combining these two meetings via a video call was suggested, she responded very positive.

*"Yes, that would be great because then I can plan and combine these meetings in time".*

*–Participant 1*

Multiple participants elaborated on why a video call would be a good alternative for the physical checkup meeting.

*"Seeing my healthcare provider makes a conversation more personal I think. And I really appreciate that."*

*–Participant 1*

*"The advantage for me would be that I can see my healthcare professional, which does provide some extra value for me."*

*–Participant 7*

To better understand the patient journey of the participants, it was discussed how they were usually notified about the telephone appointment for the checkup meeting. Many stated that the details of the telephone appointment were accessible in the patient portal or in a letter that has been sent to the patient. Several participants mentioned that a timeframe was provided in which the healthcare professional would call them to discuss the blood test results, but that this was not particularly convenient for them.

*"Next Monday, I have an appointment by telephone again and the time period for the call is between 11:00h and 13:30h, which is quite inconvenient."*

*–Participant 6*

Some of the participants said that they would have liked to see improvements on the (telephone) checkup meetings to tailor it more to their needs and wishes. An example that was given was recording the telephone call and video calls with the healthcare professional. In this way, a participant could later listen or look back at the conversation, even with their relatives.

*"Thing like that are really important in my experience, especially if you are going to suffer from dementia for example."*

*–Participant 5*

*"Yes that will work for me. If people are emotional and have to hear a story that has a lot of impact, I can imagine that not everything comes across."*

*–Participant 7*

Another suggestion that was made to help healthcare professionals in the delivery of PCC through digitalization.

*"What I have seen in another hospital is that the doctor had two screens. One for himself and one for the patient to look at, and I thought that was a very good bit of openness. Then you literally see what is being written and then the doctor does not have to turn their screen."*

*–Participant 8*

Furthermore, it was also discussed whether it was more convenient for the participants if there were broader timeslots to have an appointment in the hospital, for example after 5 p.m. Some participants stated that this could have been of added value for them.

*"I think a lot of people would benefit from being able to come after 5 p.m.. When you are able to include the patient's preferences, you really put a patient central."*

*–Participant 4*

**Self-monitoring.** Self-monitoring of one's medical condition was only applicable for patients with diabetes at the department of internal medicine. These patients could measure their own blood values and upload them digitally into the patient portal. After the values were inserted into the patient portal, patients were called by their healthcare professional who would provide advice on, for example, how much insulin to take. Participants suffering from other illnesses, like HIV and kidney failure, pointed out that self-monitoring was appealing for them as well.

*"I think this could surely be possible, especially taking my own blood pressure, weight, or something like that. I could definitely share that myself online"*

*–Participant 6*

One participant even proposed a more digitalized solution where the digital system provided an advice instead of the healthcare professional.

*"At the moment the nurse has to check my blood values and give an advice, but I think this could also be done by a system to make it somewhat faster."*

*–Participant 1*

## Post-service period

**Reviewing medical records.** In the patient portal, patients were able to review appointments, find general medical information about the medications they are taking, and make and change appointments. Not all participants made use of the possibilities of the patient's portal. It was specified that this occurred either because they were unaware of the possibilities or because they preferred other methods.

*"I know I can log in with my DigID, and I could probably see some things in there. But I actually have not really used it so far."*

*- Participant 2*

*"I have not come that far, but I know I can see my appointments in there. And that my general practitioner has a similar system."*

*–Participant 5*

All participants mentioned that they were aware that they could investigate their medical records online. However, a lack of clarity was experienced about the sharing of medical records on the patient portal and the ability of what could be seen in these digital medical records.

*"I checked my results twice on the patient portal when I was pregnant, but sometimes I could not see them."*

*–Participant 1*

*"I would like to have a better overview of every picture or scan that was once taken from me, but most of the time you can never see them after the conversation with the doctor."*

*–Participant 8*

**e-Consults.**   One participant mentioned the use of an e-Consult via the patient portal. This entailed asking a question through the portal, which was then first reviewed at the front desk and, if the assistants could not answer the question, they would send it to the doctor.

*"I do not always like calling as the person I need is not always there. I can have an e-Consult, which works very well. I send a question and it will get answered by either the front desk or the doctor. This way I do not have to wait."*

*–Participant 8*

**Online health platforms.**   The participants were asked whether they used online health platforms, like websites or Facebook pages of patient associations, for information about their chronic illness. Not all the participants made use of these platforms. Others indicated that they did look for information about the newest healthcare technologies and about other patient's experiences, but not always online.

*"I do not necessarily look into these platforms on the internet, but also in magazines. I feel that you get a lot of information from the doctor in terms of results and checks. But you have to look for the newest technologies that could improve your life for yourself. I feel that the doctors expect that."*

*–Participant 5*

*"I am a member of the patient association and sometimes I look into their Facebook page. There I see what other people experience, which is sometimes serious. Then I worry a bit which is not always helpful."*

*–Participant 4*

Table 5 presents an overview of the touchpoints that were identified in each period of the patient journey, the suggested digital alternative for each touchpoint, and which dimension(s) of PCC this digital alternative would improve, based on the dimensions identified by Gerteis et al. [4].

## Discussion

The main aim of the present study was to ascertain the chronically ill patients' needs for digital alternatives for touchpoints between them and their healthcare professionals. Specifically, we aimed to explore which digital alternatives the patients like to see implemented into their own

**Table 5. Touchpoints, alternatives, and improved PCC dimensions in the pre-service, service, and post-service period.**

| Pre-service period | (Digital) alternatives | Improved PCC dimensions |
|---|---|---|
| Intake consult | Recording the meeting | • Respect for patients' values and preferences<br>• Communication between patient and providers<br>• Involvement of family and friends<br>• Transition and continuity |
| Information leaflets | Present information through short video clips | • Respect for patients' values and preferences<br>• Communication between patient and providers |
| Check-in | Check-in screen | • Respect for patients' values and preferences<br>• Coordination of care |
| **Service period** | **(Digital) alternatives** | **Improved PCC dimensions** |
| Blood tests | Making digital appointments beforehand | • Respect for patients' values and preferences<br>• Coordination of care<br>• Communication between patient and providers |
| Checkup meetings | Getting a video call instead of physical checkup meeting | • Respect for patients' values and preferences<br>• Coordination of care<br>• Involvement of family and friends |
| Self-monitoring | Uploading results into the patient portal | • Respect for patients' values and preferences<br>• Coordination of care<br>• Communication between patient and providers |
| **Post-service period** | **(Digital) alternatives** | **Improved PCC dimensions** |
| Administrative actions | Being able to see non-medical information on a digital platform. As well as the possibility of booking e-Consults. | • Respect for patients' values and preferences<br>• Coordination of care |
| Review of medical records | Being able to see medical results and context of those results. | • Respect for patients' values and preferences |
| Contacting the hospital for questions | Being able to have e-Consults with healthcare providers. | • Respect for patients' values and preferences<br>• Coordination of care<br>• Communication between patient and providers |
| Online health platforms | Being able to see information and experiences from similar patients. | • Respect for patients' values and preferences<br>• Emotional support and alleviation of fear and anxiety |

patient journey to help healthcare professionals providing PCC. We will now discuss in more detail the findings of the current study that were listed in Table 5 with the findings from the literature.

## The patient journey of chronically ill patients

The patient journey in a hospital can be described as a three-stage process in which three different service periods are defined [13]. Though, for chronically ill patients these different periods are often not entirely applicable. The three patient groups that were included in this study are likely to be treated for their illness for the rest of their lives. Most research on the topic of patient journeys has been performed on patients that have the prospect of being cured and, therefore, leave the patient journey [36]. This implies a linear journey with a clear beginning and end. Our findings suggest that the patient journey of chronically ill patients is not a linear

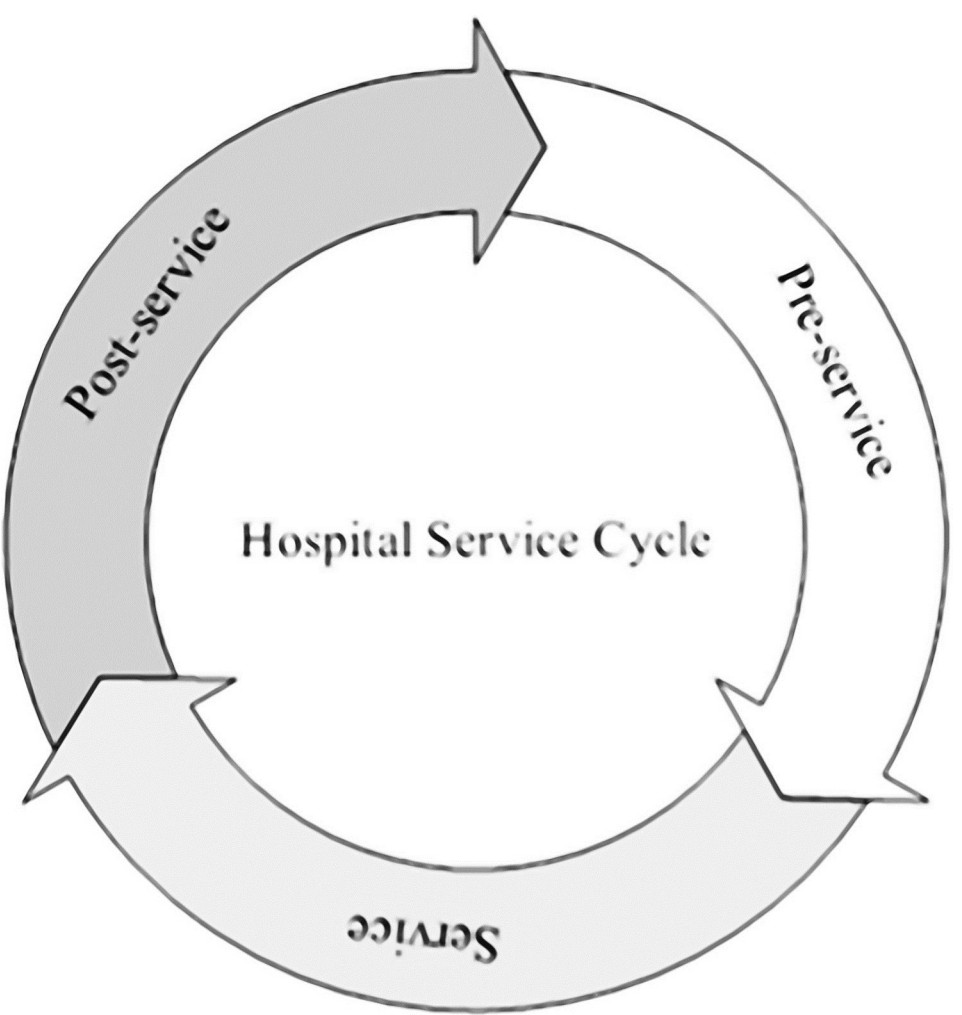

**Fig 1. Cyclical patient journey for chronically ill patients.**

journey, but a continuous cycle instead (Fig 1). For chronically ill patients it is common that a series of touchpoints happens again and again in a somewhat similar and logical sequence. In this continuous cycle, the pre-service, service, and post-service periods feed into each other. For example, a patient could perform the post-service activity of looking into the results of his blood test and use this information to ask more relevant and informed questions in a checkup meeting with the healthcare professional during the service period. This way, the journey of chronically ill patients is a continuous loop of healthcare encounters and device touchpoints.

### Touchpoints, digital alternatives, and PCC

Our findings show that the patient journey starts when a patient is referred to the hospital in accordance with previous studies [5, 19, 36]. Commonly this happens through a referral of a general practitioner or a transfer from another hospital. This start of the patient journey is usually the same for each patient, irrespective of their illness.

The check-in process in medical settings is occasionally already done through a digital interface at a screen in the waiting area to improve patient experience and to potentially reduce patient waiting time [37]. In this way, the check-in process can be perceived as more efficient

and made significantly faster when implementing this digital alternative. When adjusting the physical check-in into a digital check-in before an appointment it is no longer a healthcare encounter, but it can be considered a device touchpoint. This device touchpoint is in accordance with the majority of the patients' expressed needs and relates to a different coordination of care. Furthermore, most patients addressed their interest in a digital alternative for the information documents they receive before the intake consult takes place. By providing this information digitally, in the form of short video clips, healthcare professionals meet the majority of the patients' needs and wishes and thus provide PCC.

Regarding the service period, the most important findings were that most of the patients prefer digital check-up meetings and the option of digitally self-monitoring one's condition. During the Covid-19 pandemic, the physical check-up meetings mostly have been replaced by telephone calls. Patients perceived this as more convenient because they did not have to wait in the hospital for the results of their blood sample. Our findings suggest that a digital alternative for telephone calls, for example video calls, could also be implemented in the patient journey in order to make it more patient-centered. This is in line with the findings of a previous study [38] that found that most patients are satisfied with video calls and prefer them as an alternative to physical meetings. Patients felt more comfortable with video visits than office visits and expressed a preference for receiving future serious news via video visit, because they could be in their own supportive environment [38]. So, instead of calling patients, the check-up meeting could be done through a video call. Patients valued the option of being able to see their healthcare professional and believe that it is also valuable for the healthcare professional to see the patient. Furthermore, patients are positive about the option of recording the video call. This would make it possible for a patient to listen or look back at the conversation, even with relatives, at a convenient moment. According to the findings of this study, another digital alternative for a touchpoint in the service period that could be implemented is to make it possible for patients to monitor their own medical status. This involves that the patient uploads medical values in a digital system, which are checked by the healthcare professional who can then provide medical advice based on these values. This finding is in accordance with the eHealth Enhanced Chronic Care Model of Gee et al. [22] who propose that access to and control over personal health data due to policy changes is needed to provide the consumer more autonomy. Some health care organizations have had positive experiences with open access for consumers, including the provider notes [25].

Currently, self-monitoring is being implemented in the treatment of diabetes patients. Self-monitoring of one's medical status is a form of telemonitoring. A previous study [39] found that patients comply with telemonitoring programs and the use of technologies. Therefore, home telemonitoring of chronic diseases seems to be a promising patient management approach that produces accurate and reliable data, empowers patients, influences their attitudes and behaviors, and potentially improves their medical conditions. With technology moving at a rapid pace, non-diabetic patient groups may also benefit from self-monitoring practices in the future as well. Digitally self-monitoring was found to be very appealing for the patients, which shows their willingness to take control over their own medical process.

Concerning the post-service period, the most important finding was that most of the chronically ill patients experienced a lack of clarity about the sharing of medical records on the patient portal and the ability of what can(not) be seen in these digital files. Therefore, patients do not often use the patient portal. This finding is similar to a previous study [40] that presents that knowledge barriers on how and when to use patient portals is one of the many perceived barriers by patients to use these portals. Considering that there is a need for more knowledge and awareness about possibilities patient portals offer, it could be of added value when the healthcare professional demonstrates the possibilities and benefits of the patient portal.

Digitally providing information is also in alignment with the findings of Gee et al. [22], who argue that the role of the community to provide support for patient engagement or activation and for self-management should be expanded to include online community and health-related social networks. Moreover, healthcare professionals could also raise more awareness about e-Consults. e-Consults can be an addition to the communication channels that the hospital already makes use of, such as e-mail and telephone, which would allow the hospital to be more varied in their contact options. Additionally, it could be made possible that the healthcare professional has the option to video call the patient following the question that was asked in the e-Consult. A video call is generally perceived as a more personal way of communicating compared to a regular phone call [38]. An additional advantage of the use of e-consults is that it can reduce the waiting time for the patients that contact the hospital by phone. Patients are increasingly looking for information about their disease on their own. They are not only searching on sites supported by healthcare professionals, but also across the internet and social media like Facebook and Twitter. One of the areas where they are looking for information is on platforms where patients with the same chronic illness share their experiences. Sometimes these platforms are unregulated, but some are controlled by patient associations. The use of online health platforms could be managed by the hospital to make sure that the patient obtains reliable medical information.

## Limitations, recommendations, and future research

Clearly, there are limitations of the present study. The primary limitation of this study is the transferability of the results. Transferability entails the extent to which it can be applied in other contexts and studies [35]. Morse [35] elaborates that in qualitative research the application of the findings to another situation or population is achieved through de-contextualization and abstraction of emerging concepts and theory which should be the prerogative of the original investigator. However, in our study the findings can hardly be interpreted separate from its context. Moreover, caution must be applied with results based on a small sample size. In this study we used a qualitative, exploratory study design to develop an initial understanding of the patient journey of chronically ill patients and possible digital improvements of the touchpoints in this patient journey. We acknowledge that our sample size is smaller than recommended by the findings of multiple studies [27, 28]. The aim was to include multiple patients from four distinct patient groups: cardiovascular diseases, diabetes, HIV, and kidney failure. However, it proved to be very hard to find patients suffering from cardiovascular diseases who were willing to participate. Additionally, mainly middle-aged patients, between 35–60 years, were included and the sample size lacked patients younger than 35 years old. This could make the results of this study biased towards the middle-age group. However, previous research found that patients younger than age 65 are more likely to participate in clinical studies compared to patients 65 or older [41]. Hutchins et al. endorse this finding, as they conclude that the most consistent and largest disparity in study participation pertains to age [42]. Our findings are based on mostly middle-aged participant, but younger and older patients could potentially provide other interesting insights and suggestions of digitalization and the patient journey. Besides, according to our findings chronically ill patients have an essentially different patient journey compared to non-chronically ill patients. Our findings suggest a continuous cycle, rather than a linear patient journey, for chronically ill patients. Therefore, the results may not be entirely generalizable towards other (non-chronically ill) patient groups.

Our recommendations apply to different levels: the patient level, the healthcare professional level, and the hospital management level. On the patient level, it is advisable to keep communicating with the patients that are experiencing the touchpoints firsthand. Their feedback is

extremely valuable and often generalizable to a larger part of the patient population, as chronically ill patients often have similar wishes and needs. Additionally, there is a noteworthy number of patients that have stable conditions that would like to assess their medical status. For example, take their weight, measure blood pressure, or even blood values if possible. Improved technology could make this possible for more patients in the future, as now only diabetes patients actively perform self-monitoring activities. However, for most patients, it is not clear what digital possibilities are already available for them through the patient portal. Therefore, there is a challenge for hospitals to communicate and educate about the existing digital possibilities.

On the level of healthcare professionals, it is also important that they are aware of the existing digital possibilities. Furthermore, as is the case for patients, healthcare professionals need to be trained to use the digital platform in a consistent way. It is not preferable to have some healthcare professionals using the platform very extensively, while others are rarely making use of it.

On the hospital management level, it would be advisable to install digital check-in screens, as most of the patients do not see the added value of checking in at the desk but do so of asking questions. Consequently, employees at the desk will have more time to answer questions that patients may have. For the hospital management, this transition to digital check-in screens would also be beneficial, when keeping the increasing shortage of medical personnel in mind.

In terms of PCC, it would be desirable to perform a more extensive study of the current experiences that patients have with digitalization. Common patterns and experiences could be identified, and possible future digital applications could be based on the current experiences of the patients and their actual needs and preferences. Ideally, this would be done across the entire hospital to be able to obtain different insights from various perspectives. The concept of visualization of each interactive touchpoint that the patient experiences as they navigate the care continuum, e.g., patient journey mapping, is a relatively novel practice [36] and therefore there is no fixed or widely accepted method of establishing the patient journey. This might also be a cause for the lack of adoption of patient journey mapping in the healthcare industry [43]. Therefore, it might be interesting to capture the patient journey using a different approach, for example through questionnaires or observations or a method that solely focuses on the digital side of the patient journey. In addition, future research could build on our identified touchpoints and measure their effectiveness in relation to PCC provision [44].

## Conclusion

Literature showed that a patient journey can be divided into three distinct periods: pre-service period, service period, and post-service period. However, the empirical data showed that these periods are cyclical instead of a linear process for chronically ill patients. In this cyclical patient journey, digitalization can help put the wishes and needs of the patients at the center of care. Digital alternatives to all kinds of touchpoints within the patient journey can make the journey less of a burden and increase the level of communication between the patient and the healthcare professional. Ultimately, digitalization can help support the delivery of PCC, as it helps patients to flow more efficiently through their patient journey while establishing higher levels of communication with their healthcare professionals.

## Supporting information

**S1 Fig. Visualization of a baseline patient journey.**
(PDF)

**S1 File. A 32-item checklist for reporting qualitative studies (COREQ).**
(DOCX)

## Acknowledgments

The authors would like to thank all patients for their kind participation in the study.

## Author Contributions

**Conceptualization:** Vera K. Maas, Frederik H. Dibbets, Vincent J. T. Peters.

**Data curation:** Frederik H. Dibbets.

**Formal analysis:** Frederik H. Dibbets, Vincent J. T. Peters.

**Investigation:** Frederik H. Dibbets.

**Methodology:** Frederik H. Dibbets, Vincent J. T. Peters.

**Project administration:** Frederik H. Dibbets.

**Resources:** Daniëlle van Bijnen.

**Supervision:** Bert R. Meijboom, Daniëlle van Bijnen.

**Validation:** Vincent J. T. Peters, Bert R. Meijboom, Daniëlle van Bijnen.

**Visualization:** Vera K. Maas, Frederik H. Dibbets.

**Writing – original draft:** Vera K. Maas, Frederik H. Dibbets.

**Writing – review & editing:** Vera K. Maas, Frederik H. Dibbets, Vincent J. T. Peters, Bert R. Meijboom, Daniëlle van Bijnen.

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
