## [Decision Letter · Decision Letter 0]

10 Oct 2022

PONE-D-22-21246The never-ending patient journey of chronically ill patients: A qualitative case studyPLOS ONE

Dear Dr. Peters,

Thank you for submitting your manuscript to PLOS ONE. After careful consideration, we feel that it has merit but does not fully meet PLOS ONE’s publication criteria as it currently stands. Therefore, we invite you to submit a revised version of the manuscript that addresses the points raised during the review process.

We look forward to receiving your revised manuscript.

Kind regards,

Edward Nicol, PhD

Academic Editor

PLOS ONE

Journal Requirements:

Additional Editor Comments:

Just as observed by the reviewers, the methods can be improved. I suggest the following

**Methods**

Even though the authors mentioned in the abstract that data was collected from 8 participants, this important information is not included in the methods in main text.

Normally qualitative research requires between 9 to 24 in-depth interviews. As also highlighted by other reviewers, motivate the selection of just eight participants for the interviews; that is too little and likely to threaten the validity and generalizability of studies’ results. See the following reference for ideal sample size.   Vasileiou, K., Barnett, J., Thorpe, S. *et al.* Characterising and justifying sample size sufficiency in interview-based studies: systematic analysis of qualitative health research over a 15-year period. *BMC Med Res Methodol* **18**, 148 (2018). https://doi.org/10.1186/s12874-018-0594-7Monique Hennink, Bonnie N. Kaiser, Sample sizes for saturation in qualitative research: A systematic review of empirical tests, Social Science & Medicine, Volume 292, 2022, https://doi.org/10.1016/j.socscimed.2021.114523.
As observed by the reviewer, semi-structured interview is not always the best method for exploratory study design. Unstructured interviews such as in-depth interviews would have been the most appropriate for of data collection method for your study.

Your ethics statement should come after the data analysis section, just before the results section. You should also include the ethic reference number

Reviewers' comments:

Reviewer's Responses to Questions

**Comments to the Author**

1. Is the manuscript technically sound, and do the data support the conclusions?

Reviewer #1: No

Reviewer #2: Partly

Reviewer #3: Yes

2. Has the statistical analysis been performed appropriately and rigorously? 

Reviewer #1: N/A

Reviewer #2: N/A

Reviewer #3: N/A

3. Have the authors made all data underlying the findings in their manuscript fully available?

Reviewer #1: No

Reviewer #2: No

Reviewer #3: Yes

4. Is the manuscript presented in an intelligible fashion and written in standard English?

Reviewer #1: Yes

Reviewer #2: No

Reviewer #3: Yes

5. Review Comments to the Author

Reviewer #1: Thank you for this opportunity to review your manuscript in which you describe the patient journey of chronically ill patients as cyclical and never-ending and could be augmented by digitalisation of touchpoints. Overall, this is interesting research, and your study could help inform policy and clinical practice. Your paper could be improved in the following ways.

Background.

The literature is well-used to support your research question. A lot of what you talk about in terms of patient-centric care and digitalisation can be related to the Chronic Care Model (1) in which the touchpoints you refer to can be considered ‘productive interactions’. Did you have a reason for not using this model in your research? If so, I suspect you will need to justify why you didn’t use the model. In my opinion its use gives strength to a lot of what you did.

Methods.

You say you used purposive sampling but what you describe sounds more like convenience sampling, e.g., you had access to certain departments in a hospital and those departments covered certain health issues which in turn influenced who responded to your invitation to participate in interviews. In other words, you did interviews with people who were available (convenience) rather than your first choice of interviews (purposive). Please consider modifying your description of the type of sampling you did.

I would prefer to see you create a table of the interview questions and insert it in the body of the manuscript than for you to have it as supplementary information. That way a time-poor reader doesn’t have to follow links to interview questions. Also, please consider adding the demographic items in the table so that it’s clear when and how you got that information, i.e., that you didn’t get demographic data from a third party (the clinic).

There are certain qualities in qualitative data, especially in interviews, that need to be unpacked in the methods section, e.g., data saturation, why you only completed eight interviews and what satisfied you that you could stop at eight, what constituted your case study (you say it was a case study but nowhere do you say why it was a case study and how interviews within a case study are useful to answer your research question). Take a look at this article by Morse (2) about ways to strengthen qualitative research so that you can indicate what you did to mitigate the weaknesses/limitations of a set of interviews. You may also find this article useful to explain your decision to interview the eight people you interviewed and the role of the interviewees’ context (3). You need to describe your methods decisions in such a way that another researcher can replicate your methods. At the moment, I’m not confident that your study can be replicated.(4)

I think that the coding table should go into the methods section with a brief statement about how you derived the codes.

Findings.

You indicate that the demographic profile of the interviewees is skewed to older people in this section and in the limitations. Is it possible for you to get a report on the demographics of the clinic patients who form the context of your research? All you need to get is a demographic profile of ages from the three clinics from which you sourced your interviews to be able to say if self-selection for interviews was because people with those conditions are usually older or older people tend to self-select for research participation. There is some literature to support the latter conclusion, and it shouldn’t be hard to find, assuming that is what you discovered in your findings.

There are too many quotes. This leaves the reader feeling that they need to do some analysis – the analysis is your job. The use of multiple quotes for one point creates questions in the reader’s mind – did you finish coding the quotes, were these the only quotes for this point, does this mean that if you used one quote for a point that only one person gave you a quote for that point?

Table 2 about touchpoints belongs in the findings section. It’s a very nice table and can be used to complement the thematic narrative.

I like the figure that demonstrates the cyclic nature of the touchpoints, but it’s not clear in the findings that you saw this. Placement of this figure into the findings section (not the discussion section) should be reconsidered.

Because you can't make the raw data freely available you should consider not using supplementary files and rather insert the content of your supplementary files into the manuscript where they below.

Discussion

There are some very good descriptions of the findings in the discussion section. Some of these descriptions belong on the findings section. A good discussion should (1) summarise the key findings (done, but repetitive), (2) compare and contrast the findings with what’s already in the literature (partially done), (3) indicate how the research question/s was/were answered (could be done more clearly). It is in this section that you can draw strongly on the Chronic Care Model (and the associated literature about digitisation of ‘productive interactions’) to strengthen the points you want to make from your findings.

Limitations.

You can’t use quantitative research measures to critique the value and truth of qualitative research. It is more appropriate to talk about transferability, trustworthiness, data saturation, triangulation and member describing the strengths and weaknesses of your research methods. Use Morse’s article to help you do that. Use Malterud’s article to help justify the number of interviews you did. Indicate what types of contexts can and should use your research findings, i.e., similar to the context of your research.

References.

The list of references appears to be focused well on the research being reported, and articles from a range of years are used. I have referred below to articles that you should use to strengthen your manuscript.

General proofreading issues.

There is a tendency to use the odd word oddly, e.g.,

- In the abstract you say, ‘These digital alternatives consisted of video … viewing their own medical status in a digital manner’ Do you mean ‘digital format’? The former is ambiguous.

- When the subject of a sentence is singular/plural you make the rest of the sentence plural/singular, e.g., ‘Each patient is unique and requires treatments that suits …’ Suits should be suit because at the subject of the sentence is ‘each patient’ implies plural. Confusing I know. I recommend that you get someone to go through the manuscript and clean these minor errors up for you.

- People can’t be ‘it’, e.g., ‘Some of these healthcare encounters that are currently taking place physically at the hospital could also be performed by the patient itself.’ Delete ‘itself’.

My references for this review:

1. Gee PM, Greenwood DA, Paterniti DA, Ward D, Miller LMS. The eHealth enhanced chronic care model: a theory derivation approach. Journal of medical Internet research. 2015;17(4).

2. Morse JM. Critical analysis of strategies for determining rigor in qualitative inquiry. Qualitative health research. 2015;25(9):1212-22.

3. Malterud K, Siersma VD, Guassora AD. Sample size in qualitative interview studies: guided by information power. Qualitative health research. 2016;26(13):1753-60.

4. Coiera E, Ammenwerth E, Georgiou A, Magrabi FJJotAMIA. Does health informatics have a replication crisis? 2018;25(8):963-8.

Reviewer #2: These are some comments to improve the current study

Major revision of the title - based on the intent - multiple points of contact and PCC with physical /face to face and technology.

Basically this study is about whether integrating digital technology in the continuity and continuum of care would lead to a quality outcomes.

The study is more like pilot /feasibility study.

Figure 1- Revisit and align with the intent of the study, that is the continuum of care -disagree that it is a cyclical process it is rather a non-linear process -depending on their condition and the aspect of self-monitoring need to be somehow factored in.

Sup File 1 - The "stop" in the figure contradicts the cyclical process/ continuum -see also the result section which goes beyond the health system.

In the research design/method section -Appears to be more towards a survey method using structured interview approach and does not qualify for a qualitative research with phenomenological method of data analysis.

A descriptive exploratory research design using a survey method through structured interviews is evident with the listed questionnaire items in the supplementary file.

Provide some detail on the reliability and validity / trustworthiness, etc. irrespective of research design and methods used.

Lastly, the concept of consumerism with touch points in their journey lean towards a concrete rather than abstract construct which weakly align with qualitative approach.

Reviewer #3: The manuscript “the never-ending patient journey of chronically ill patients: A qualitative case study” is of great importance and interest. The manuscript has some serious originality concerns as the similarity index is 49% which is exceptionally high. The authors have made no such disclosures if it is part of some previous project or extracted from the dissertation. Without such information, it is difficult for me to comment on the other aspects of the manuscript. Therefore, I recommend the rejection of the manuscript at this stage.

6. PLOS authors have the option to publish the peer review history of their article (what does this mean?). If published, this will include your full peer review and any attached files.

Reviewer #1: **Yes: **Karen Day

Reviewer #2: **Yes: **Chandra R Makanjee

Reviewer #3: **Yes: **Imran Hameed Khaliq

---

## [Author Response · Author response to Decision Letter 0]

24 Nov 2022

Journal requirements

Q1 Please ensure that your manuscript meets PLOS ONE's style requirements, including those for file naming. The PLOS ONE style templates can be found at

R: We have carefully checked the PLOS ONE’s style requirements, including those for file naming.

Q2 In your Data Availability statement, you have not specified where the minimal data set underlying the results described in your manuscript can be found. PLOS defines a study's minimal data set as the underlying data used to reach the conclusions drawn in the manuscript and any additional data required to replicate the reported study findings in their entirety. All PLOS journals require that the minimal data set be made fully available. For more information about our data policy, please see http://journals.plos.org/plosone/s/data-availability.

R: We have now specified in the cover letter where the minimal data set underlying the results described in our manuscript can be found. Thanks for updating our Data Availability statement accordingly.

 

Editor

Q3. Just as observed by the reviewers, the methods can be improved. I suggest the following

R: Thank you for your comments. We will go over your suggestions point-by-point below.

Methods

Q4. Even though the authors mentioned in the abstract that data was collected from 8 participants, this important information is not included in the methods in main text.

R: We have now included the number of participants in our manuscript.

C: The data were collected through eight semi-structured interviews. (Page 11).

Q5. Normally qualitative research requires between 9 to 24 in-depth interviews. As also highlighted by other reviewers, motivate the selection of just eight participants for the interviews; that is too little and likely to threaten the validity and generalizability of studies’ results. See the following reference for ideal sample size. 

Vasileiou, K., Barnett, J., Thorpe, S. et al. Characterising and justifying sample size sufficiency in interview-based studies: systematic analysis of qualitative health research over a 15-year period. BMC Med Res Methodol 18, 148 (2018). https://doi.org/10.1186/s12874-018-0594-7

Monique Hennink, Bonnie N. Kaiser, Sample sizes for saturation in qualitative research: A systematic review of empirical tests, Social Science & Medicine, Volume 292, 2022, https://doi.org/10.1016/j.socscimed.2021.114523.

R: We agree that the number of respondents of our study is relatively low. However, given the fact that we conducted an exploratory study, previous studies have shown that a limited amount of interviews can be sufficient in order to get a reliable sense of thematic exhaustion and variability. We encountered data saturation after 7 interviews, as no new themes or touchpoints emerged from the data gathered between interview seven and interview eight. We incorporated one of the suggested references and an additional reference to elaborate on our reasoning and justification for interviewing eight participants. Please see our manuscript accordingly.

C: The sample size aim was to include three patients of each of the patient groups. This number was based on research that proves that data saturation can be achieved with 9 to 17 interviews [25]. Generally, reaching saturation, meaning new interviews do not yield new data on the interview topics, is considered sufficient for validity [26]. In the case of exploratory studies, a limited amount of interviews can be sufficient [26] in order to get a reliable sense of thematic exhaustion and variability within our data set. (Page 10). 

Additional references:

25. Hennink M, Kaiser BN. Sample sizes for saturation in qualitative research: A systematic review of empirical tests. Social Science & Medicine. 2021;292: 114523. https://doi.org/10.1016/j.socscimed.2021.114523.

26. Guest G, Bunce A, Johnson L. How many interviews are enough? An experiment with data saturation and variability. Field Methods. 2006;18(1): 59-82.

https://doi.org/10.1177/1525822X05279903. 

Q6. As observed by the reviewer, semi-structured interview is not always the best method for exploratory study design. Unstructured interviews such as in-depth interviews would have been the most appropriate for of data collection method for your study.

R: We agree with the reviewer that a semi-structured interview is not always the best method for exploratory study design. We have added a reference to strengthen our reasoning for the use of semi-structured interviews and, for next studies, will carefully consider the use of unstructured interviews for exploratory study design.

C: A semi-structured interview lends itself well to qualitative research, as it is an efficient way to gather rich data and it allows for the creation of new insights into the studied case [27]. (Page 11). 

Additional reference:

27. Verleye K. Designing, writing-up and reviewing case study research: An equifinality perspective. Journal of Service Management. 2019;30(5): 549-576.

https://doi.org/10.1108/JOSM-08-2019-0257. 

Q7. Your ethics statement should come after the data analysis section, just before the results section. You should also include the ethic reference number

R: We have amended the order of the Methods section; the ethical considerations section now follows the validity and reliability section. Furthermore, we have added the ethic reference number. Please see our manuscript accordingly.

C. The Ethics Review Board of Catharina Hospital Eindhoven approved our study design (nWMO-2021.055). (Page 15).

Reviewer #1

Q8. Thank you for this opportunity to review your manuscript in which you describe the patient journey of chronically ill patients as cyclical and never-ending and could be augmented by digitalisation of touchpoints. Overall, this is interesting research, and your study could help inform policy and clinical practice. Your paper could be improved in the following ways.

R: We thank the reviewer for the compliments and will go over her comments point-by-point below.

Background.

Q9. The literature is well-used to support your research question. A lot of what you talk about in terms of patient-centric care and digitalisation can be related to the Chronic Care Model (1) in which the touchpoints you refer to can be considered ‘productive interactions’. Did you have a reason for not using this model in your research? If so, I suspect you will need to justify why you didn’t use the model. In my opinion its use gives strength to a lot of what you did.

R: We aimed to explore which digital alternatives patients would like to see implemented into their patient journey to help healthcare professionals providing patient-centered care. We have considered using the Chronic Care Model in the design of the study, but belief our focus is more in alignment with the seven dimensions affecting PCC as introduced by Gerteis et al. (1993). We now acknowledge the existence of the Chronic Care Model in the manuscript.

C: Although we acknowledge the eHealth Enhanced Chronic Care Model, as proposed by Gee et al. [22], which places chronic care in the context of the community where the person will receive care, we are convinced that the work by Gerteis et al. [4] provides a more comprehensive view on how digitalization can improve dimensions of PCC. (Page 8).

Additional reference:

22. Gee PM, Greenwood DA, Paterniti DA, Ward D, Miller LMS. The eHealth enhanced chronic care model: A theory derivation approach. Journal of medical Internet research. 2015;17(4): 4-5.

https://doi:10.2196/jmir.4067. 

Methods.

Q10. You say you used purposive sampling but what you describe sounds more like convenience sampling, e.g., you had access to certain departments in a hospital and those departments covered certain health issues which in turn influenced who responded to your invitation to participate in interviews. In other words, you did interviews with people who were available (convenience) rather than your first choice of interviews (purposive). Please consider modifying your description of the type of sampling you did.

R: We agree that convenience sampling technique is a better description of our sampling strategy. We modified the manuscript accordingly. 

C: Recruitment of these participants was carried out based on convenience sampling logic. (Page 10).

Q11. I would prefer to see you create a table of the interview questions and insert it in the body of the manuscript than for you to have it as supplementary information. That way a time-poor reader doesn’t have to follow links to interview questions. Also, please consider adding the demographic items in the table so that it’s clear when and how you got that information, i.e., that you didn’t get demographic data from a third party (the clinic).

R: We agree with the reviewer that it is better to include the interview questions into the manuscript itself as opposed to providing it as a supplementary file. Please see our manuscript accordingly (Page 11 and 12).

Q12. There are certain qualities in qualitative data, especially in interviews, that need to be unpacked in the methods section, e.g., data saturation, why you only completed eight interviews and what satisfied you that you could stop at eight, what constituted your case study (you say it was a case study but nowhere do you say why it was a case study and how interviews within a case study are useful to answer your research question). Take a look at this article by Morse (2) about ways to strengthen qualitative research so that you can indicate what you did to mitigate the weaknesses/limitations of a set of interviews. You may also find this article useful to explain your decision to interview the eight people you interviewed and the role of the interviewees’ context (3). You need to describe your methods decisions in such a way that another researcher can replicate your methods. At the moment, I’m not confident that your study can be replicated.(4)

R: We now include more detail on data saturation, why we only completed eight interviews, and what constituted our case study. Please see our manuscript accordingly. 

C: The sample size aim was to include three patients of each of the patient groups. This number was based on research that proves that data saturation can be achieved with 9 to 17 interviews [25]. Generally, reaching saturation, meaning new interviews do not yield new data on the interview topics, is considered sufficient for validity [26]. In the case of exploratory studies, a limited amount of interviews can be sufficient [26] in order to get a reliable sense of thematic exhaustion and variability within our data set. (Page 10).

Additional references:

25. Hennink M, Kaiser BN. Sample sizes for saturation in qualitative research: A systematic review of empirical tests. Social Science & Medicine. 2021;292: 114523. https://doi.org/10.1016/j.socscimed.2021.114523.

26. Guest G, Bunce A, Johnson L. How many interviews are enough? An experiment with data saturation and variability. Field Methods. 2006;18(1): 59-82.

https://doi.org/10.1177/1525822X05279903. 

Q13. I think that the coding table should go into the methods section with a brief statement about how you derived the codes.

R: We have added the coding table into the data analysis section of the Methods section with a brief statement on how we derived the codes. Please see the manuscript accordingly. (Page 15, 16 and 17).

C: The coding framework was continuously discussed and tested during the coding of the interviews, which is in alignment with the main principle that Miles et al. endorse and entails that codes should ‘have some conceptual and structural unity. Codes should relate to one another in coherent, study-important ways; they should be part of a unified structure’. (Page 13).

Findings.

Q14. You indicate that the demographic profile of the interviewees is skewed to older people in this section and in the limitations. Is it possible for you to get a report on the demographics of the clinic patients who form the context of your research? All you need to get is a demographic profile of ages from the three clinics from which you sourced your interviews to be able to say if self-selection for interviews was because people with those conditions are usually older or older people tend to self-select for research participation. There is some literature to support the latter conclusion, and it shouldn’t be hard to find, assuming that is what you discovered in your findings.

R: We thank the reviewer for the suggestion to add a demographic profile of ages of the patient groups to support our statement that the age of the interviewees is skewed to middle-aged patients. We have added this information in the form of Table 1 (Demographic profile of ages) on page 10 and 11. We incorporated two additional references to substantiate our statement. Please see the manuscript accordingly. 

C: Table 1 provides an overview of the demographic profile of these four patient groups. (page 10).

Table 1. Demographic profile of patient groups at department of Internal Medicine. (page 10).

Patient group Total number of patients (N) Age (M, SD) Age (Min-Max)

Arteriosclerosis 12 62.0 (11.65) 42-76

Diabetes 1535 58.4 (22.29) 17-95

HIV 615 47.4 (18.45) 19-88

Kidney failure 598 71.6 (20.84) 19-97

However, Saphner et al. [37] found that patients younger than age 65 are more likely to participate in clinical studies compared to patients 65 or older. Hutchins et al. endorse this finding, as they conclude that the most consistent and largest disparity in study participation pertains to age [38]. Our findings are based on mostly middle-aged participant, but younger and older patients could potentially provide other interesting insights and suggestions of digitalization and the patient journey. (Page 31).

Additional references:

37. Saphner T, Marek A, Homa JK, Robinson L, Glandt N. Clinical trial participation assessed by age, sex, race, ethnicity, and socioeconomic status. Contemp Clin Trials. 2021 Apr;103: 106315. 

https://doi: 10.1016/j.cct.2021.106315. 

38. Hutchins LF, Unger JM, Crowley JJ, Coltman CA Jr, Albain KS. Underrepresentation of patients 65 years of age or older in cancer-treatment trials. N Engl J Med. 1999 Dec 30;341(27): 2061-7. 

https://doi: 10.1056/NEJM199912303412706.

Q15. There are too many quotes. This leaves the reader feeling that they need to do some analysis – the analysis is your job. The use of multiple quotes for one point creates questions in the reader’s mind – did you finish coding the quotes, were these the only quotes for this point, does this mean that if you used one quote for a point that only one person gave you a quote for that point?

R: We agree with the reviewer that we have made abundant use of quotes to illustrate our findings. We have carefully read our Results section and have reduced the number of quotes. Initially there were 42 quotes in the Results section, in the revised manuscript we kept 31 quotes. Please see our manuscript accordingly.

Q16. Table 2 about touchpoints belongs in the findings section. It’s a very nice table and can be used to complement the thematic narrative.

R: We thank the reviewer for the compliments on the table about the touchpoints, which we use to substantiate and clarify the link between our empirics and the literature. We therefore believe that it is placed appropriately in the Discussion section. 

Q17. I like the figure that demonstrates the cyclic nature of the touchpoints, but it’s not clear in the findings that you saw this. Placement of this figure into the findings section (not the discussion section) should be reconsidered.

R: We believe that Figure 1 comes into its own in the Discussion section, because this figure was created on the basis of our interpretation of the study results rather than a readily available internal process. In the Results sections, we do not want to elaborate on interpreting the results, we want to leave that for the Discussion section. 

Q18. Because you can't make the raw data freely available you should consider not using supplementary files and rather insert the content of your supplementary files into the manuscript where they below.

R: Thank you for sharing these thoughts with us. We have decided to upload the minimal data set to replicate the study to a stable, public repository. In addition, we have included the content of the supplementary files into the manuscript where they belong when appropriate. Please see the manuscript accordingly. 

Discussion

Q19. There are some very good descriptions of the findings in the discussion section. Some of these descriptions belong on the findings section. A good discussion should (1) summarise the key findings (done, but repetitive), (2) compare and contrast the findings with what’s already in the literature (partially done), (3) indicate how the research question/s was/were answered (could be done more clearly). It is in this section that you can draw strongly on the Chronic Care Model (and the associated literature about digitisation of ‘productive interactions’) to strengthen the points you want to make from your findings.

R: Thank you for your suggestions. We have now rewritten (and removed) some parts of the discussion. We now also build on the Chronic Care Model as per your suggestion. Please see the manuscript accordingly.

C: This finding is in accordance with the eHealth Enhanced Chronic Care Model of Gee et al. who propose that access to and control over personal health data due to policy changes is needed to provide the consumer more autonomy. Some health care organizations have had positive experiences with open access for consumers, including the provider notes [22]. (Page 28-29).

Digitally providing information is also in alignment with the findings of Gee et al. [22], who argue that the role of the community to provide support for patient engagement or activation and for self-management should be expanded to include online community and health-related social networks. (Page 29-30).

Additional reference:

22. Gee PM, Greenwood DA, Paterniti DA, Ward D, Miller LMS. The eHealth enhanced chronic care model: A theory derivation approach. Journal of medical Internet research. 2015;17(4): 4-5.

https://doi:10.2196/jmir.4067. 

Limitations.

Q20. You can’t use quantitative research measures to critique the value and truth of qualitative research. It is more appropriate to talk about transferability, trustworthiness, data saturation, triangulation and member describing the strengths and weaknesses of your research methods. Use Morse’s article to help you do that. Use Malterud’s article to help justify the number of interviews you did. Indicate what types of contexts can and should use your research findings, i.e., similar to the context of your research.

R: We did not intend to only use quantitative research measures to critique our qualitative work. We have been more careful in our wording and build our argument on the article of Morse.

C: Transferability entails the extent to which it can be applied in other contexts and studies [31]. Morse et al. [31] elaborates that in qualitative research the application of the findings to another situation or population is achieved through de-contextualization and abstraction of emerging concepts and theory which should be the prerogative of the original investigator. However, in our study the findings can hardly be interpreted separate from its context. Moreover, caution must be applied with results based on a small sample size. (Page 30-31).

Additional reference:

31. Morse JM. Critical analysis of strategies for determining rigor in qualitative inquiry. Qualitative Health Research. 2015;25(9): 1212-1222.

https://doi.org/10.1177/1049732315588501

References.

Q21. The list of references appears to be focused well on the research being reported, and articles from a range of years are used. I have referred below to articles that you should use to strengthen your manuscript.

R: We thank the reviewer for her suggested references and have incorporated those to strengthen our manuscript. Please see the manuscript accordingly.

General proofreading issues.

Q22. There is a tendency to use the odd word oddly, e.g., In the abstract you say, ‘These digital alternatives consisted of video … viewing their own medical status in a digital manner’ Do you mean ‘digital format’? The former is ambiguous.

R: We agree with the reviewer and changed the wording accordingly. We also changed this elsewhere in the manuscript.

C: These digital alternatives consisted of video calls, digitally checking in before a physical appointment, digitally self-monitoring one’s medical condition and personally uploading monitoring results into the patient portal, and viewing their own medical status in a digital format. (Page 3).

Therefore, it might be interesting to capture the patient journey using a different approach. (Page 32).

Q23. When the subject of a sentence is singular/plural you make the rest of the sentence plural/singular, e.g., ‘Each patient is unique and requires treatments that suits …’ Suits should be suit because at the subject of the sentence is ‘each patient’ implies plural. Confusing I know. I recommend that you get someone to go through the manuscript and clean these minor errors up for you.

R: We agree with the reviewer that we have been sloppy in terms of singular/plural wording. We cleaned these minor errors up with the help of an English native speaker.

C: Each patient is unique and requires treatments that suit their situation in a patient-centered way. (Page 5).

Q24. People can’t be ‘it’, e.g., ‘Some of these healthcare encounters that are currently taking place physically at the hospital could also be performed by the patient itself.’ Delete ‘itself’.

R: We fully agree that people cannot be referred to as it. We have carefully read the manuscript and have deleted those kind of references. Please see the manuscript accordingly.

Q25. My references for this review:

1. Gee PM, Greenwood DA, Paterniti DA, Ward D, Miller LMS. The eHealth enhanced chronic care model: a theory derivation approach. Journal of medical Internet research. 2015;17(4).

2. Morse JM. Critical analysis of strategies for determining rigor in qualitative inquiry. Qualitative health research. 2015;25(9):1212-22.

3. Malterud K, Siersma VD, Guassora AD. Sample size in qualitative interview studies: guided by information power. Qualitative health research. 2016;26(13):1753-60.

4. Coiera E, Ammenwerth E, Georgiou A, Magrabi FJJotAMIA. Does health informatics have a replication crisis? 2018;25(8):963-8.

R: We thank the reviewer for the suggested references and have incorporated those, when appropriate, in our manuscript.

Reviewer #2

Q26. These are some comments to improve the current study

R: We thank the reviewer for the comments to improve our manuscript and will go over the comments point-by-point below.

Q27. Major revision of the title – based on the intent – multiple points of contact and PCC with physical /face to face and technology.

R: We agree with the reviewer that the title of our manuscript could be more clearly tailored to the intent of our study, and therefore we updated it. Please see the manuscript accordingly. 

C: The never-ending patient journey of chronically ill patients: A qualitative case study on touchpoints in relation to patient-centered care (Page 1 and 2).

Q28. Basically this study is about whether integrating digital technology in the continuity and continuum of care would lead to a quality outcomes. The study is more like pilot /feasibility study.

R: This study indeed can be viewed more like a pilot/feasibility study about whether integrating digital technology in the continuity and continuum of care can lead to quality outcomes for chronically ill patients. We have changed some wording in the manuscript to highlight this. 

Q29. Figure 1- Revisit and align with the intent of the study, that is the continuum of care -disagree that it is a cyclical process it is rather a non-linear process -depending on their condition and the aspect of self-monitoring need to be somehow factored in.

R: There is certainly something to be said that the patient journey of chronically ill patients can be described as a non-linear process. However, when describing something as non-linear it means that it does not progress or develop from one stage tot the next stage in a logical way. This is not the case in our study, because the patient actually does progress from one stage to another. Instead, we argue that the patient journey of chronically ill patients is a cyclical process in which a series of events happens again and again in a somewhat similar and logical sequence. In the Discussion section we briefly address this. Please see the manuscript accordingly. 

C: For chronically ill patients it is common that a series of touchpoints happens again and again in a somewhat similar and logical sequence. (Page 25).

Q30. Sup File 1 - The "stop" in the figure contradicts the cyclical process/ continuum -see also the result section which goes beyond the health system.

R: The ‘stop’ in figure S1 “Visualization of the baseline patient journey” refers to the baseline patient journey of a patient in a hospital. This journey entails a clear beginning and end (as the patient is cured), and, therefore, there is a ‘stop’ in this figure. The cyclical continuum in Figure 1 refers to the patient journey of chronically ill patients. Since they are likely to be treated for the rest of their lives, their patient journey does not entail a clear ending. Additionally, in the patient journey of chronically ill patients events tend to happen again and again in a somewhat similar sequence. Without this clear ending and with these recurring events, we argue that the patient journey of this specific patient group is cyclical instead of linear. 

Q31. In the research design/method section -Appears to be more towards a survey method using structured interview approach and does not qualify for a qualitative research with phenomenological method of data analysis.

R: The applied study method is a qualitative, exploratory case study approach. The consolidated criteria for reporting qualitative research (COREQ) were used as guidelines for the study design and the data analysis. We actually do believe that it therefore qualifies for a qualitative research with phenomenological method of data analysis.

Q32. A descriptive exploratory research design using a survey method through structured interviews is evident with the listed questionnaire items in the supplementary file.

R: The listed interview items in the supplementary file are now added to the main manuscript (Page 11 and 12). These questions were asked during the interviews, however not in a fixed order which is common practice in semi-structured interviews. We used these questions as a guideline for our interviews, but we also asked follow-up questions in relation to the answers of the participants, per the nature of semi-structured interviews. Consequently, we made use of a semi-structured interview design. 

Q33. Provide some detail on the reliability and validity / trustworthiness, etc. irrespective of research design and methods used.

R: We thank the reviewer for this suggestion and agree that we should provide some details on these topics. That is why we now have added a section about validity and reliability into the Methods section in which we elaborate on these topics. This new section is placed between the data analysis section and the ethical considerations section. Additionally we elaborate a bit on the topic of transferability in the Discussion section. 

C: To establish validity and reliability of our data, several measures have been taken [30]. To improve the internal validity, the interviewed patients were initially contacted by their healthcare professionals to ask whether they would take part in this research. The study and its purpose were explained. To further increase reliability, the semi-structured interviews have been conducted in Dutch, the native language of the participants, so that no language barriers exist. Additionally the member checking method was applied. This method entails to check that the transcription was correctly done by returning the transcript of the interview to the participants. [31]. The transcripts were returned to the eight participants. We received no comments and corrections. (Page 15).

C: Transferability entails the extent to which it can be applied in other contexts and studies [31]. Morse et al. [31] elaborates that in qualitative research the application of the findings to another situation or population is achieved through de-contextualization and abstraction of emerging concepts and theory which should be the prerogative of the original investigator. However, in our study the findings can hardly be interpreted separate from its context. Moreover, caution must be applied with results based on a small sample size. (Page 30 and 31).

Additional references:

30. Golafshani, N. Understanding reliability and validity in qualitative research. The Qualitative Report. 2003:8(4): 597-606. 

https://doi.org/10.46743/2160-3715/2003.1870.

31. Morse JM. Critical analysis of strategies for determining rigor in qualitative inquiry. Qualitative Health Research. 2015;25(9): 1212-1222.

https://doi.org/10.1177/1049732315588501.

Q34. Lastly, the concept of consumerism with touch points in their journey leans towards a concrete rather than abstract construct which weakly align with qualitative approach.

R: We understand the concern of the reviewer that in the context of consumerism touchpoints can be considered concrete, perhaps even physical constructs rather than abstract constructs. However, speaking of a “patient journey’ as a metaphor is abstract in itself and given the fact that it was not clear which touchpoints were present in the patient journey, we first had to conduct an exploratory, qualitative approach to identify the touchpoints present in this ‘journey’. Future research can build on our study and use a quantitative approach to further uncover the potential of touch points.

C: Therefore, it might be interesting to capture the patient journey using a different approach, for example through questionnaires or observations or a method that solely focuses on the digital side of the patient journey. In addition, future research could build on our identified touchpoints and measure their effectiveness in relation to PCC provision. (Page 33 and 33).

Reviewer #3

Q35. The manuscript “the never-ending patient journey of chronically ill patients: A qualitative case study” is of great importance and interest. The manuscript has some serious originality concerns as the similarity index is 49% which is exceptionally high. The authors have made no such disclosures if it is part of some previous project or extracted from the dissertation. Without such information, it is difficult for me to comment on the other aspects of the manuscript. Therefore, I recommend the rejection of the manuscript at this stage.

R: Thank you for acknowledging the importance of our manuscript. We have carefully looked at the similarity index and acknowledge that 40% of our content stems from the Master’s thesis. Part of the study has been published as a Master thesis. We have the approval to republish the data from the University where the Master thesis has been written. We also cite the source in our article. We belief this should not be an issue, since this is original work by one of the authors of this study. We hope that by clarifying the comment raised, and highlighting this in the cover letter, you will be able to comment on the other aspects of our manuscript.

Additional reference:

40. Dibbets, FH. The digital side of healthcare: How digitalizing the patient journey can increase patient-centered care. [master’s thesis]. Tilburg (NL): Tilburg University; 2022.

---

## [Decision Letter · Decision Letter 1]

18 Jan 2023

PONE-D-22-21246R1

The never-ending patient journey of chronically ill patients: A qualitative case study on touchpoints in relation to patient-centered care

PLOS ONE

Dear Dr. Peters,

Thank you for submitting your manuscript to PLOS ONE. After careful consideration, we feel that it has merit but does not fully meet PLOS ONE’s publication criteria as it currently stands. Therefore, we invite you to submit a revised version of the manuscript that addresses the points raised during the review process.

A major concern for two of the three reviewers were issues related to the methodology used in this study, and I concur with their comments. This section should be clearly written, with a clear description of each of the sub-sections for ease of replication. The appropriate approach for this study is a semi-structured in-depth interview and this should pull through in your data collection and analysis sections. Actually what the authors described in the data collection section is a face-to-face (or virtual) individual in-depth interviews which were audio recorded.

Other issue raised was the adequacy of the eight purposive samples used in this study, for which the authors responded with the argument of reaching saturation and cited* *Guest et al., 2006. How many interviews are enough? An experiment with data saturation and variability.

However, the excerpt below from this paper says otherwise *"Although the idea of saturation is helpful at the conceptual level, it provides little practical guidance for estimating sample sizes, prior to data collection, necessary for conducting quality research."  "*Purposive samples still need to be carefully selected, and twelve interviews will likely not be enough if a selected group is relatively heterogeneous, the data quality is poor, and the domain of inquiry is diffuse and/or vague. Likewise, you will need larger samples if your goal is to assess variation between distinct groups or correlation among variables. For most research enterprises, however, in which the aim is to understand common perceptions and experiences among a group of relatively homogeneous individuals, twelve interviews should suffice.*" *Also refer to:

Monique Hennink, Bonnie N. Kaiser, Sample sizes for saturation in qualitative research: A systematic review of empirical tests, Social Science & Medicine, Volume 292, 2022, https://doi.org/10.1016/j.socscimed.2021.114523

Vasileiou, K., Barnett, J., Thorpe, S. *et al.* Characterising and justifying sample size sufficiency in interview-based studies: systematic analysis of qualitative health research over a 15-year period. *BMC Med Res Methodol* 18, 148 (2018). https://doi.org/10.1186/s12874-018-0594-7

Reviewer #1 made very useful comments that were not adequately addressed. see the following:

*"Table 2 about touchpoints belongs in the findings section. It’s a very nice table and can be used to complement the thematic narrative."  *I suggest including Table 2 in your result section and discuss it in further in your the discussion section. 

"There are certain qualities in qualitative data, especially in interviews, that need to be unpacked in the methods section, e.g.,... what constituted your case study (you say it was a case study but nowhere do you say why it was a case study and how interviews within a case study are useful to answer your research question). ...You need to describe your methods decisions in such a way that another researcher can replicate your methods. At the moment, I’m not confident that your study can be replicated.(4)" 

"...did you finish coding the quotes, were these the only quotes for this point, does this mean that if you used one quote for a point that only one person gave you a quote for that point?"

Even though the authors claimed to have used thematic analysis, it is unclear what approach was used for the analysis. Deductive or inductive approach, or both? 

Refer to the following paper on how to conduct a case study:

Crowe, S., Cresswell, K., Robertson, A. *et al.* The case study approach. *BMC Med Res Methodol* 11, 100 (2011). https://doi.org/10.1186/1471-2288-11-100

The authors were given a chance to rectify these flaws, however, these have not been adequately addressed in the revised submission. For these reasons, I cannot recommend this manuscript for publication in the current form.

We look forward to receiving your revised manuscript.

Kind regards,

Edward Nicol, PhD

Academic Editor

PLOS ONE

Journal Requirements:

Reviewers' comments:

Reviewer's Responses to Questions

**Comments to the Author**

1. If the authors have adequately addressed your comments raised in a previous round of review and you feel that this manuscript is now acceptable for publication, you may indicate that here to bypass the “Comments to the Author” section, enter your conflict of interest statement in the “Confidential to Editor” section, and submit your "Accept" recommendation.

Reviewer #2: (No Response)

Reviewer #3: All comments have been addressed

2. Is the manuscript technically sound, and do the data support the conclusions?

Reviewer #2: No

Reviewer #3: Yes

3. Has the statistical analysis been performed appropriately and rigorously? 

Reviewer #2: N/A

Reviewer #3: N/A

4. Have the authors made all data underlying the findings in their manuscript fully available?

Reviewer #2: No

Reviewer #3: Yes

5. Is the manuscript presented in an intelligible fashion and written in standard English?

Reviewer #2: No

Reviewer #3: Yes

6. Review Comments to the Author

Reviewer #2: though the topic is sound and relevant. Currently the study has methodological flaws and require a careful thought through approach whether it qualify for a qualitative approach.

Reviewer #3: The Authors performed adequate revisions and the scientific contribution of the work has improved. As a result of this, I recommend accepting the manuscript for publication in PLOS One.

7. PLOS authors have the option to publish the peer review history of their article (what does this mean?). If published, this will include your full peer review and any attached files.

Reviewer #2: No

Reviewer #3: **Yes: **Imran Hameed Khaliq

---

## [Author Response · Author response to Decision Letter 1]

16 Feb 2023

Editor

Q1. A major concern for two of the three reviewers were issues related to the methodology used in this study, and I concur with their comments. This section should be clearly written, with a clear description of each of the sub-sections for ease of replication. The appropriate approach for this study is a semi-structured in-depth interview and this should pull through in your data collection and analysis sections. Actually what the authors described in the data collection section is a face-to-face (or virtual) individual in-depth interviews which were audio recorded.

R: Thank you for your comments. We agree that the Methods section should be clearly written for ease of replication. We have conducted a major overhaul of the Methods section for improvements in terms of transparency and replication purposes. Please see the relevant changes for this comment below. 

C. A qualitative, exploratory case study design was used to identify the touchpoints between chronically ill patients and healthcare professionals during the patient journey and to explore opportunities for digital alternatives for these touchpoints to make the patient journey more patient-centered. Because the topic of this study is in its formative stage, we conducted qualitative research in the form of a case study. Case study research designs have different purposes, namely exploratory, explanatory, and descriptive, or a combination of those purposes [23]. The design of the current study can be defined as exploratory, as it explores the patient journey of chronically ill patients, their attitudes towards that journey, and the possible digital alternatives provided by these patients for the encountered touchpoints during their journey. We used a case study approach to generate an in-depth understanding of a complex issue in its real-life context, which lends itself well to capture information on ‘how’, ‘what’, and, ‘why’ questions [24]. An advantage of conducting research in this manner is that it provides the opportunity to study a topic in its real-life context; this can contribute to understanding whether digital alternatives for touchpoints make the patient journey of chronically ill patients more patient-centered. The consolidated criteria for reporting qualitative research (COREQ) [25] were used as guidelines for the study design and the data analysis (S1 File). (Page 9 and 10).

The study data were gathered using a cross-sectional approach, meaning that we collected data from different individuals at a single point in time. The data were collected through eight semi-structured interviews. The semi-structured nature of the interviews allowed us to make sure that important topics were covered, while leaving room for the participants to provide their story regarding their patient journey [30]. A semi-structured interview is relevant for exploratory research, as it is an effective way to gather rich data and it allows for the creation of new insights into the case under study [23]. As a consequence of Covid-19 restrictions in the Netherlands, some of these interviews were conducted via Zoom instead of face-to face. We also collected patient characteristics such as age, gender, type of disease and how long the participants were living with their chronic illness. The interview questions were grouped in advance according to the three phases of the patient journey: pre-service period, service period, and post-service period [13]. A case study approach is particularly useful when there is a need to obtain an in-depth appreciation of an issue, in which qualitative interviews are preferred over quantitative methods alone [24]. Therefore, we are convinced that our data collection approach is aligned with the aim of our study. In each interview the same questions were asked, but not in a fixed order (Table 2). The semi-structured approach was used to obtain insights in the touchpoints in the patient journey during different periods. In addition, we collected participants’ needs, preferences, and suggestions for digital alternatives for the touchpoints per period of the patient journey. The interviews were audio recorded and transcribed verbatim.

We also collected relevant documentation that was open to the public (e.g. information leaflets, invitation letters), and internal documentation of the department of internal medicine (e.g. planning schemes, medical protocols). These collected documents provided valuable information in terms of touchpoints encountered by chronically ill patients during their patient journey. (Page 13).

The data were analyzed using the three steps method for thematic analysis as described by Miles et al. [31]: 1) data reduction, 2) data display, and 3) drawing conclusions. This is a systematic data reduction process building on, among others, the reading of transcripts, document summaries, codification of text segments, generation of themes and categories, and identification of relationships [31]. We employed this deductive, thematic analysis approach since we used a coding framework for analysis that was based on concepts and definitions derived from the literature. While exploratory case studies probe into and shed light on what is essentially unknown, they should be guided by a specific purpose that frames the research [23]. The deductive codes were useful in both the segmentation and coding phase of the data analysis. Three main categories were created to store the coded data: touchpoints, patient-centered care, and digital solutions. The definitions of these categories and their subcategories can be found in Table 3. These (sub)categories were later used to label the data with the relevant codes. The coding framework was continuously discussed and tested during the coding of the interviews, which is in alignment with the main principle that Miles et al. [31] endorse and entails that codes should ‘have some conceptual and structural unity. Codes should relate to one another in coherent, study-important ways; they should be part of a unified structure’ [31]. 

We started with developing a baseline patient journey, which is common practice in patient journey research [32], based on the data commonalities that were obtained from the interviews (S1 Fig). Next, as our guiding principle, we used the touchpoint definition in customer journeys of Lemon & Verhoef [6] to identify touchpoints in the patient journey: a moment of contact between a firm or service provider and a customer at distinct points in the customer journey (p. 71). It was then discussed within the research team whether these touchpoints could be described as healthcare encounters or device touchpoints. To obtain more detailed information about the touchpoints and about the moments at which they occurred in the patient journey, the touchpoints were grouped together according to the three periods of the patient journey, as identified by Rosenbaum et al. [13]. After the coding process, we systematically analyzed the touchpoints to explore whether digital alternatives that were provided by the participants could be used in the patient journey and how this could improve PCC, using the dimensions identified by Gerteis et al. [4]. (Page 16-17).

Additional references:

23. Verleye K. Designing, writing-up and reviewing case study research: An equifinality perspective. Journal of Service Management. 2019;30(5): 549-576. doi: 10.1108/JOSM-08-2019-0257.

24. Crowe S, Cresswell K, Robertson A, Huby G, Avery A, Sheikh A. The case study approach. BMC Medical Research Methodology. 2011;11(100): 1-9. doi: 10.1186/147-2288-11-100.

Q2. Other issue raised was the adequacy of the eight purposive samples used in this study, for which the authors responded with the argument of reaching saturation and cited Guest et al., 2006. How many interviews are enough? An experiment with data saturation and variability.

However, the excerpt below from this paper says otherwise "Although the idea of saturation is helpful at the conceptual level, it provides little practical guidance for estimating sample sizes, prior to data collection, necessary for conducting quality research." "Purposive samples still need to be carefully selected, and twelve interviews will likely not be enough if a selected group is relatively heterogeneous, the data quality is poor, and the domain of inquiry is diffuse and/or vague. Likewise, you will need larger samples if your goal is to assess variation between distinct groups or correlation among variables. For most research enterprises, however, in which the aim is to understand common perceptions and experiences among a group of relatively homogeneous individuals, twelve interviews should suffice." Also refer to:

Monique Hennink, Bonnie N. Kaiser, Sample sizes for saturation in qualitative research: A systematic review of empirical tests, Social Science & Medicine, Volume 292, 2022, https://doi.org/10.1016/j.socscimed.2021.114523

Vasileiou, K., Barnett, J., Thorpe, S. et al. Characterising and justifying sample size sufficiency in interview-based studies: systematic analysis of qualitative health research over a 15-year period. BMC Med Res Methodol 18, 148 (2018). https://doi.org/10.1186/s12874-018-0594-7

R: We have conducted a major overhaul of the subsection ‘Participants’ of the Methods section to describe the adequacy of the eight purposive samples used in this study in more detail. We have included the suggested references in our manuscript. Please see the manuscript accordingly.

C. Chronically ill patients who are treated and monitored at the department of internal medicine were asked to participate. Recruitment of these participants was carried out based on convenience sampling logic [26]. To capture a wide variety of experiences of patients who are being treated at the department of internal medicine, we focused on recruiting patients who suffer from kidney disease (kidney failure), infectious disease (HIV), vascular diseases (arteriosclerosis), and metabolic diseases (diabetes). Table 1 provides an overview of the demographic profile of ages of these four patient groups. Potential participants were given as much time as needed to consider whether they wished to participate and, in the case of a positive decision, were asked to reply to the internist and give consent for their contact details to be disclosed to the study team. The participants who gave consent were then contacted by telephone to schedule the interview. The sample size aim was to include at least three patients of each patient group, so that we would conduct twelve interviews in total. This number of interviews was based on previous research that proves that data saturation, meaning new interviews do not yield new data on the interview topics, can be achieved with 9 to 17 interviews [27, 28]. Eight patients agreed to participate in this study, despite our continued efforts to include more patients. Although the number of participants is relatively small, a limited amount of interviews can be sufficient in the case of exploratory studies in order to get a reliable sense of thematic exhaustion and variability of the data [27, 28]. In addition, research has indicated that sample size should not be considered alone but be embedded in the more encompassing examination of data adequacy [29]. This implies that sample size numbers in qualitative research are not unimportant, but should be extended to terms of adequate amounts of evidence, adequate variety in kinds of evidence, and adequate interpretive status of evidence. Given our comprehensive data collection, consisting of semi-structured interviews and document analysis, we are convinced that, despite our relatively small number of interviews, we were able to present adequate variety and interpretation of our evidence. By combining the information from the interviews and document analysis with the theoretical framework, we were able to draw a comprehensive image of the patient journey for the patient groups included in our study. For instance, when an interviewee mentioned a specific touchpoint, we verified this touchpoint with our collected documents and interview transcripts to ensure that this touchpoints was in accordance with the baseline patient journey we developed. This approach strengthened us in our belief that we were able to present convincing evidence. This conviction also arose during the data analysis when we observed that data saturation happened after seven interviews, as no new themes emerged from the data gathered between interview seven and interview eight. This is in accordance with previous research that demonstrated that saturation can be achieved in a narrow range of interviews [27]. (Page 10, 11 and 12).

In this study we used a qualitative, exploratory study design to develop an initial understanding of the patient journey of chronically ill patients and possible digital improvements of the touchpoints in this patient journey. We acknowledge that our sample size is smaller than recommended by the findings of multiple studies [27, 28]. The aim was to include multiple patients from four distinct patient groups: HIV, diabetes, kidney failure, and cardiovascular diseases. However, it proved to be very hard to find patients suffering from cardiovascular diseases who were willing to participate. (Page 36 and 37).

Additional references:

27. Hennink M, Kaiser BN. Sample sizes for saturation in qualitative research: A systematic review of empirical tests. Social Science & Medicine. 2021;292: 114523. doi: 10.1016/j.socscimed.2021.114523.

29. Vasileiou K, Barnett J, Thorpe S, Young T. Characterising and justifying sample size sufficiency in interview-based studies: Systematic analysis of qualitative health research over a 15-year period. BMC Medical Research Methodology. 2018;18(148): doi: 10.1186/s12874-018-0594-7. 

Q3. Reviewer #1 made very useful comments that were not adequately addressed. See the following: “Table 2 about touchpoints belongs in the findings section. It’s a very nice table and can be used to complement the thematic narrative." I suggest including Table 2 in your result section and discuss it in further in your discussion section. 

R: We apologize for not adequately addressing the useful comments of reviewer #1 about the placement of Table 2. In our initial manuscript, this was Table 2. In our first revised manuscript it was labeled as Table 5, though it was still placed in the Discussion section. We now have placed Table 5 at the end of the Results section and discuss it in further detail in our discussion section. Please see our manuscript accordingly. 

C. Table 5 presents an overview of the touchpoints we identified in each period of the patient journey, the suggested digital alternative for each touchpoint, and which dimension(s) of PCC this digital alternative would improve, based on the dimensions identified by Gerteis et al. [4]. (Page 29). 

We will now discuss in more detail the findings of the current study that were listed in Table 5 with the findings from the literature. (Page 32).

Q4. "There are certain qualities in qualitative data, especially in interviews, that need to be unpacked in the methods section, e.g.,... what constituted your case study (you say it was a case study but nowhere do you say why it was a case study and how interviews within a case study are useful to answer your research question). ...You need to describe your methods decisions in such a way that another researcher can replicate your methods. At the moment, I’m not confident that your study can be replicated.(4)" 

Refer to the following paper on how to conduct a case study:

Crowe, S., Cresswell, K., Robertson, A. et al. The case study approach. BMC Med Res Methodol 11, 100 (2011). https://doi.org/10.1186/1471-2288-11-100

R. We conducted a major overhaul of our Methods section and provide more detail on what constituted our case study, why it was a case study, and how interviews were useful to answer our research question. These adjustments will add to the ease of replication of our study. We have used the suggested reference and included it in the reference list.

C. A qualitative, exploratory case study design was used to identify the touchpoints between chronically ill patients and healthcare professionals during the patient journey and to explore opportunities for digital alternatives for these touchpoints to make the patient journey more patient-centered. Because the topic of this study is in its formative stage, we conducted qualitative research in the form of a case study. Case study research designs have different purposes, namely exploratory, explanatory, and descriptive, or a combination of those purposes [23]. The design of the current study can be defined as exploratory, as it explores the patient journey of chronically ill patients, their attitudes towards that journey, and the possible digital alternatives provided by these patients for the encountered touchpoints during their journey. We used a case study approach to generate an in-depth understanding of a complex issue in its real-life context, which lends itself well to capture information on ‘how’, ‘what’, and, ‘why’ questions [24]. An advantage of conducting research in this manner is that it provides the opportunity to study a topic in its real-life context; this can contribute to understanding whether digital alternatives for touchpoints make the patient journey of chronically ill patients more patient-centered. (Page 9 and 10).

We took the healthcare provision for chronically ill patients characterized by continuous, monitoring episodes, provided at the department of internal medicine of a hospital in the south of the Netherlands, as our case. The department consists of outpatient clinics for patients who suffer from (chronic) illnesses such as kidney diseases, infectious diseases, vascular diseases, and metabolic diseases. The case sampling of the participants focused on typical cases of care for chronically ill patients characterized by continuous, monitoring episodes at this department. In the fall of 2021, using e-mail and telephone, the study team approached three internists at the department of internal medicine to help us select and contact potential participants. Potential participants were considered for inclusion if they visited the department of internal medicine and are treated for either kidney failure, HIV, arteriosclerosis, or diabetes. (Page 10).

The study data were gathered using a cross-sectional approach, meaning that we collected data from different individuals at a single point in time. The data were collected through eight semi-structured interviews. The semi-structured nature of the interviews allowed us to make sure that important topics were covered, while leaving room for the participants to provide their story regarding their patient journey [30]. A semi-structured interview is relevant for exploratory research, as it is an effective way to gather rich data and it allows for the creation of new insights into the case under study [23]. As a consequence of Covid-19 restrictions in the Netherlands, some of these interviews were conducted via Zoom instead of face-to face. We also collected patient characteristics such as age, gender, type of disease and how long the participants were living with their chronic illness. The interview questions were grouped in advance according to the three phases of the patient journey: pre-service period, service period, and post-service period [13]. A case study approach is particularly useful when there is a need to obtain an in-depth appreciation of an issue, in which qualitative interviews are preferred over quantitative methods alone [24]. Therefore, we are convinced that our data collection approach is aligned with the aim of our study. In each interview the same questions were asked, but not in a fixed order (Table 2). The semi-structured approach was used to obtain insights in the touchpoints in the patient journey during different periods. (Page 12 and 13).

Additional reference:

24. Crowe S, Cresswell K, Robertson A, Huby G, Avery A, Sheikh A. The case study approach. BMC Medical Research Methodology. 2011;11(100): 1-9. doi: 10.1186/147-2288-11-100.

30. Saunders M, Lewis P, Thornhill A. Research methods for business students. Harlow: Pearson Education; 2009.

Q5. "...did you finish coding the quotes, were these the only quotes for this point, does this mean that if you used one quote for a point that only one person gave you a quote for that point?"

R. When we used a single quote in our Results section, this does not indicate that only one participant gave a quote for that point. We then used this quote because, in our opinion, this quote illustrated and clarified the findings of the interviews. In response to the previous comments made by Reviewer #1 in the first round of revision, we removed 11 quotes. Initially, there were 42 quotes in the Results section, in the previous revised manuscript we kept 31 quotes to illustrate our findings. 

Q6. Even though the authors claimed to have used thematic analysis, it is unclear what approach was used for the analysis. Deductive or inductive approach, or both? 

R. We have elaborated on the approach we used for the thematic analysis in the subsection ‘Data analysis’ of the Methods section. Please see our manuscript accordingly. 

C. The data were analyzed using the three steps method for thematic analysis as described by Miles et al. [31]: 1) data reduction, 2) data display, and 3) drawing conclusions. This is a systematic data reduction process building on, among others, the reading of transcripts, document summaries, codification of text segments, generation of themes and categories, and identification of relationships [31]. We employed this deductive, thematic analysis approach since we used a coding framework for analysis that was based on concepts and definitions derived from the literature. While exploratory case studies probe into and shed light on what is essentially unknown, they should be guided by a specific purpose that frames the research [23]. The deductive codes were useful in both the segmentation and coding phase of the data analysis. (Page 16).

Additional reference:

23. Verleye K. Designing, writing-up and reviewing case study research: An equifinality perspective. Journal of Service Management. 2019;30(5): 549-576. doi: 10.1108/JOSM-08-2019-0257.

Q7. The authors were given a chance to rectify these flaws, however, these have not been adequately addressed in the revised submission. For these reasons, I cannot recommend this manuscript for publication in the current form.

R. We apologize for not adequately rectifying the flaws in our revised submission. We hope that the improvements we have currently made make the manuscript now suitable for publication. 

Q8. Please review your reference list to ensure that it is complete and correct. If you have cited papers that have been retracted, please include the rationale for doing so in the manuscript text, or remove these references and replace them with relevant current references. Any changes to the reference list should be mentioned in the rebuttal letter that accompanies your revised manuscript. If you need to cite a retracted article, indicate the article’s retracted status in the References list and also include a citation and full reference for the retraction notice.

R. We have now carefully reviewed our reference list to ensure that it is complete and correct. We have included additional references based on the suggestions by the editor and the reviewer(s). Please see the manuscript accordingly.

Reviewer #2

Q9. Though the topic is sound and relevant. Currently the study has methodological flaws and require a careful thought through approach whether it qualify for a qualitative approach.

R. We thank reviewer #2 for acknowledging that the topic of our manuscript is sound and relevant. We hope that by addressing the comments of the editor we also have improved the methodological flaws and comments concerning the qualitative approach mentioned by Reviewer #2. 

Reviewer #3

Q10. The Authors performed adequate revisions and the scientific contribution of the work has improved. As a result of this, I recommend accepting the manuscript for publication in PLOS One.

R: We thank reviewer #3 for acknowledging that we made adequate revisions and for recommending to accept our manuscript for publication in PLoS ONE.

---

## [Decision Letter · Decision Letter 2]

22 Mar 2023

PONE-D-22-21246R2The never-ending patient journey of chronically ill patients: A qualitative case study on touchpoints in relation to patient-centered carePLOS ONE

Dear Dr. Peters,

Thank you for submitting your manuscript to PLOS ONE. After careful consideration, we feel that it has merit but does not fully meet PLOS ONE’s publication criteria as it currently stands. Therefore, we invite you to submit a revised version of the manuscript that addresses the points raised during the review process. To ensure the Reviewers will be able to recommend that your revised manuscript is accepted, please pay careful attention to each of the comments that have been pasted underneath this email. This way we can avoid future rounds of clarifications and revisions, moving swiftly to a decision.

We look forward to receiving your revised manuscript.

Kind regards,

Edward Nicol, PhD

Academic Editor

PLOS ONE

Journal Requirements:

Reviewers' comments:

Reviewer's Responses to Questions

**Comments to the Author**

1. If the authors have adequately addressed your comments raised in a previous round of review and you feel that this manuscript is now acceptable for publication, you may indicate that here to bypass the “Comments to the Author” section, enter your conflict of interest statement in the “Confidential to Editor” section, and submit your "Accept" recommendation.

Reviewer #3: (No Response)

Reviewer #4: All comments have been addressed

2. Is the manuscript technically sound, and do the data support the conclusions?

Reviewer #3: Partly

Reviewer #4: Yes

3. Has the statistical analysis been performed appropriately and rigorously? 

Reviewer #3: N/A

Reviewer #4: N/A

4. Have the authors made all data underlying the findings in their manuscript fully available?

Reviewer #3: Yes

Reviewer #4: Yes

5. Is the manuscript presented in an intelligible fashion and written in standard English?

Reviewer #3: No

Reviewer #4: Yes

6. Review Comments to the Author

Reviewer #3: I have reviewed your manuscript and have compiled a list of major revisions based on the suggestions provided:

Page 3: Line 41-44: Authors may consider rephrasing this sentence for more clarity: "Participants were included if they had visited the department of internal medicine and had received treatment for either diabetes, kidney failure, arteriosclerosis, or HIV."

Page 10: Line 218: I would like to suggest a small change to the section on the sampling method.

In the manuscript, the authors have referred to the recruitment method as "convenience sampling logic". I believe that this may cause some confusion among readers. It would be more accurate to simply state that convenience sampling was used to select participants and that purposive sampling was used to target specific chronic illnesses in order to capture a wide variety of experiences.

I think this clarification would make the sampling method more understandable to readers and help address any potential confusion that may arise.

Page 11: Line 222: In the participant recruitment section, the authors mentioned that potential participants were given as much time as needed to consider whether they wished to participate. However, it would be helpful to readers if you could clarify the specific time frame given to participants to decide.

I would suggest adding a sentence to this section that specifies the amount of time given to participants, for example, "Participants were given one week to consider whether they wished to participate in the study." This would provide readers with a clear understanding of the time frame used in the recruitment process.

Page 12: Line 252: In the current presentation of the table, the headings and the order of the columns are as follows: "Patient Group, number of patients, Average age, Standard deviation, minimum age, maximum age". I would like to suggest reordering the columns and changing the heading "Patient Group" to "Patient Group by Disease" for improved clarity. Additionally, it would be helpful if you could shuffle the data according to the following sequence:

"Number of patients, Patient Group by Disease, Average age, Standard deviation, minimum age, maximum age"

Finally, I would suggest placing the table in the section of the manuscript where it is being referred to, for easier reference by the readers.

In the current presentation, the figures have been reported with varying numbers of decimal places. For example, some figures have been reported with one decimal place, while others have been reported with two decimal places. This inconsistency may cause confusion for readers.

To improve the clarity and consistency of your manuscript, I suggest using a consistent style of reporting figures. Specifically, I recommend reporting all figures with two decimal places to ensure consistency and clarity throughout the manuscript.

Page 12: Line 255: Given the qualitative and exploratory nature of the study, the focus was primarily on understanding the experiences, perceptions, and attitudes of individuals rather than on collecting quantitative data at a single point in time. Therefore, instead of mentioning the cross-sectional approach, the authors may want to emphasize the qualitative and exploratory aspects of the study, and how these informed the data collection and analysis methods used. This can provide readers with a clearer understanding of the purpose and scope of the study.

Page 12: Line 262: Given that Zoom is a widely-used video conferencing platform, it may be helpful for readers to have a more specific reference to the company and location in the manuscript. Therefore, we suggest that the authors include the company's name, Zoom Video Communications, Inc., and its headquarters location, San Jose, California, USA, when referring to the video conferencing software used in the study. This can enhance the clarity and transparency of the methods section for readers.

Page 13: Line 266: I noticed that the authors mentioned the benefits of a case study approach in both the previous section and this section. It may be beneficial to merge similar information in one place to avoid repeating information.

Page 13: Line 268: While it is important to align the data collection approach with the aim of the study, the reader can draw their own conclusion from the reported data. Therefore, omitting the statement in the conclusion section may be more appropriate.

Page 13: Line 270: I have noticed that the phrase "the semi-structured approach" appears multiple times in the methods section of your manuscript. While it is important to describe the methodology thoroughly, I believe that some of the repetitions may be unnecessary and could disrupt the text's flow.

Therefore, I suggest you review the methods section and consider consolidating some of the descriptions or finding alternative ways to convey the same information without repeating the exact same phrase multiple times.

Page 13: Line 280: It may be beneficial to split the question "What is your age and gender?" into two separate questions to improve the clarity of the data collected.

Page 16: Line 289: Add references here of that literature here.

Page 16: Line 296: Is this text referencing style in line with PLoS One house style?

Page 20: Line 316: Consider rephrasing "have been" to "were"

Page 20: Line 323: The authors mentioned that the transcripts were returned to all participants, and no comments or corrections were received. Can you kindly provide some additional information on the timeline for this process? Specifically, how long did you wait for participants to review and return the transcripts before considering it as no comments received? This information will help readers understand the study's timeline and the data verification process.

Page 20: Line 327: Can you please clarify the decision made by the Ethics Review Board regarding the no necessity of Medical Research Involving Human Subject (WMO) approval from a Medical Ethics Committee? It would be helpful to understand the reasons behind this decision in order to ensure that ethical considerations were thoroughly evaluated and followed in the study.

Page 20, 21: Line 334 - 341: I have reviewed the Results section and would like to recommend restructuring this section. Specifically, I believe the information about the number of patients approached, and reasons for non-participation is more appropriate in the Methods section than in the Results section.

As per scientific writing conventions, the Results section should focus on presenting the study's key findings, while the Methods section provides details about the study design, recruitment process, and other pertinent information. Therefore, I would suggest revising the Results section to focus solely on the presentation of the interview results, as well as the touchpoints and digital alternatives explored in the study.

Page 20: Line 336: In the results section, please present the interview findings in past tense since the study has already been conducted and you are reporting the results.

Page 21: Line 346 and 347: It is recommended to report the results using the appropriate scientific notation, such as the use of symbols and punctuation marks. For instance, instead of using a comma to separate the mean and standard deviation, it is more appropriate to use a semi-colon to separate them, as in M=50; SD=13.09. This notation is widely used in scientific writing and can help to ensure clarity and consistency in the reporting of results.

Overall, these revisions will help improve the clarity and consistency of your manuscript, and ensure that readers can understand the study's methods, findings, and conclusions more easily.

Reviewer #4: (No Response)

7. PLOS authors have the option to publish the peer review history of their article (what does this mean?). If published, this will include your full peer review and any attached files.

Reviewer #3: **Yes: **Imran Hameed Khaliq

Reviewer #4: No

---

## [Author Response · Author response to Decision Letter 2]

13 Apr 2023

Reviewer 3

Q1. I have reviewed your manuscript and have compiled a list of major revisions based on the suggestions provided.

R. We thank the reviewer for his extensive comments. We have addressed your comments in more detail below.

Q2. Page 3: Line 41-44: Authors may consider rephrasing this sentence for more clarity: "Participants were included if they had visited the department of internal medicine and had received treatment for either diabetes, kidney failure, arteriosclerosis, or HIV."

R. We thank the reviewer for this suggestion and have now rephrased the sentence for more clarity. In addition, we now also present the chronic illnesses in alphabetical order throughout the manuscript.

C. Participants were included if they had visited the department of internal medicine and had received treatment for either arteriosclerosis, diabetes, HIV, or kidney failure. (Page 3).

Q3. Page 10: Line 218: I would like to suggest a small change to the section on the sampling method.

In the manuscript, the authors have referred to the recruitment method as "convenience sampling logic". I believe that this may cause some confusion among readers. It would be more accurate to simply state that convenience sampling was used to select participants and that purposive sampling was used to target specific chronic illnesses in order to capture a wide variety of experiences.

I think this clarification would make the sampling method more understandable to readers and help address any potential confusion that may arise.

R. We thank the reviewer for this suggestion and agree that this clarification can prevent any potential confusion that may arise. We have changed the manuscript accordingly.

C. Chronically ill patients who received treatment and are monitored at the department of internal medicine were asked to participate. First, convenience sampling logic was used to identify potential participants [26]. Second, purposive sampling was used to target specific chronic illnesses to capture a wide variety of experiences of patients who are being treated at the department of internal medicine, with a focus on patients who suffered from infectious disease (HIV), kidney disease (kidney failure), metabolic diseases (diabetes), and vascular diseases (arteriosclerosis). (Page 10 and 11).

Q4. Page 11: Line 222: In the participant recruitment section, the authors mentioned that potential participants were given as much time as needed to consider whether they wished to participate. However, it would be helpful to readers if you could clarify the specific time frame given to participants to decide.

I would suggest adding a sentence to this section that specifies the amount of time given to participants, for example, "Participants were given one week to consider whether they wished to participate in the study." This would provide readers with a clear understanding of the time frame used in the recruitment process.

R. We have now added more detail to specify the amount of time given to participants to cinsider whether they wished to participate. In the Netherlands, two weeks is considered an appropriate time frame to consider potential participation in scientific research. Please see the manuscript accordingly.

C. Potential participants were given two weeks to consider whether they wished to participate and, in the case of a positive decision, were asked to reply to the internist and give consent for their contact details to be disclosed to the study team. (Page 11).

Q5. Page 12: Line 252: In the current presentation of the table, the headings and the order of the columns are as follows: "Patient Group, number of patients, Average age, Standard deviation, minimum age, maximum age". I would like to suggest reordering the columns and changing the heading "Patient Group" to "Patient Group by Disease" for improved clarity. Additionally, it would be helpful if you could shuffle the data according to the following sequence:

"Number of patients, Patient Group by Disease, Average age, Standard deviation, minimum age, maximum age"

Finally, I would suggest placing the table in the section of the manuscript where it is being referred to, for easier reference by the readers.

R. We have changed the order of the data of Table 1 according to the suggested sequence. Please see the manuscript accordingly.

C. 

Number of patients Patient group by disease Average age Standard deviation Minimum age Maximum age

12 Arteriosclerosis 62.01 11.65 42 76

1535 Diabetes 58.43 22.29 17 95

615 HIV 47.75 18.45 19 88

598 Kidney failure 71.62 20.84 19 97

(Page 12).

Q6. In the current presentation, the figures have been reported with varying numbers of decimal places. For example, some figures have been reported with one decimal place, while others have been reported with two decimal places. This inconsistency may cause confusion for readers. To improve the clarity and consistency of your manuscript, I suggest using a consistent style of reporting figures. Specifically, I recommend reporting all figures with two decimal places to ensure consistency and clarity throughout the manuscript.

R. We agree with the reviewer that we have been inconsistent in terms of decimal places. We have now ensured that we consistently apply two decimal places throughout the manuscript. 

C. The study team approached twelve chronically ill patients of the department of internal medicine. Eight of them agreed to participate in the present study, none of them were from the cardiovascular patient group. Reasons for not participating that were given were no interest in participation or not having sufficient time for the interview. Table 2 provides an overview of the participant characteristics. Three of the participants were male and five of them were female. The youngest participant was 38 years old and the oldest was 78 years old (M=50.00 years; SD=13.09). On average, participants were living with their chronic illness for 18 years at the time of the interviews (M=18.00 years; SD=11.26). The interviews lasted from 20 to 55 minutes. (Page 12).

Q7. Page 12: Line 255: Given the qualitative and exploratory nature of the study, the focus was primarily on understanding the experiences, perceptions, and attitudes of individuals rather than on collecting quantitative data at a single point in time. Therefore, instead of mentioning the cross-sectional approach, the authors may want to emphasize the qualitative and exploratory aspects of the study, and how these informed the data collection and analysis methods used. This can provide readers with a clearer understanding of the purpose and scope of the study.

R. We agree that emphasizing the qualitative and exploratory aspects provides the readers with a clearer understanding of the purpose and scope of our study. 

C. We conducted eight semi-structured interviews which allowed us to make sure that important topics were covered, while leaving room for the participants to provide their story regarding their patient journey [30]. Given the qualitative and exploratory nature of our case study, the focus was primarily on understanding the experiences, perceptions, and attitudes of chronically ill patients during their patient journey. The semi-structured interview approach is relevant for exploratory research, as it is an effective way to gather rich data and it allows for the creation of new insights into the case under study [23]. (Page 13).

Q8. Page 12: Line 262: Given that Zoom is a widely-used video conferencing platform, it may be helpful for readers to have a more specific reference to the company and location in the manuscript. Therefore, we suggest that the authors include the company's name, Zoom Video Communications, Inc., and its headquarters location, San Jose, California, USA, when referring to the video conferencing software used in the study. This can enhance the clarity and transparency of the methods section for readers.

R. We thank the reviewer for this suggestion and now have added a more specific reference in the manuscript.

C. As a consequence of Covid-19 restrictions in the Netherlands in 2021, five interviews were conducted via Zoom [31] instead of face-to face. (Page 13).

Additional reference:

31. Zoom Video Communications, Inc. Zoom. Version 5.8.3 [software]. San Jose: Zoom Video Communications, Inc; 2021. Available from: https://zoom.us/

Q9. Page 13: Line 266: I noticed that the authors mentioned the benefits of a case study approach in both the previous section and this section. It may be beneficial to merge similar information in one place to avoid repeating information.

R. To avoid repeating the information related to the benefits of a case study approach we grouped these statements together in the Study design paragraph. Please see our manuscript accordingly. 

C. A qualitative, exploratory case study design was used to identify the touchpoints between chronically ill patients and healthcare professionals during the patient journey and to explore opportunities for digital alternatives for these touchpoints to make the patient journey more patient-centered. Because the topic of this study is in its formative stage, we conducted qualitative research in the form of a case study. Case study research designs have different purposes, namely exploratory, explanatory, and descriptive, or a combination of those purposes [23]. The design of the current study can be defined as exploratory, as it explores the patient journey of chronically ill patients, their attitudes towards that journey, and the possible digital alternatives provided by these patients for the encountered touchpoints during their journey. We used a case study approach to generate an in-depth understanding of a complex issue in its real-life context, which lends itself well to capture information on ‘how’, ‘what’, and ‘why’ questions [24]. An advantage of conducting research in this manner is that it provides the opportunity to study a topic in its real-life context; this can contribute to understanding whether digital alternatives for touchpoints make the patient journey of chronically ill patients more patient-centered. The consolidated criteria for reporting qualitative research (COREQ) [25] were used as guidelines for the study design and the data analysis (S1 File). (Page 9 and 10).

Q10. Page 13: Line 268: While it is important to align the data collection approach with the aim of the study, the reader can draw their own conclusion from the reported data. Therefore, omitting the statement in the conclusion section may be more appropriate.

R. We have now omitted the statement to ensure that readers can draw their own conclusion from the reported data. Please see the manuscript accordingly.

Q11. Page 13: Line 270: I have noticed that the phrase "the semi-structured approach" appears multiple times in the methods section of your manuscript. While it is important to describe the methodology thoroughly, I believe that some of the repetitions may be unnecessary and could disrupt the text's flow. Therefore, I suggest you review the methods section and consider consolidating some of the descriptions or finding alternative ways to convey the same information without repeating the exact same phrase multiple times.

R. We agree with the reviewer that we have used the phrase “the semi-structured approach” too often. We have carefully reviewed the Methods section and have removed some of the repetition and rewritten sentences to convey the same message. Please see the Methods section accordingly. 

C. Given our comprehensive data collection, consisting of semi-structured interviews and document analysis, we are convinced that, despite our relatively small number of interviews, we were able to present adequate variety and interpretation of our evidence. (Page 11).

We conducted eight semi-structured interviews which allowed us to make sure that important topics were covered, while leaving room for the participants to provide their story regarding their patient journey [30]. Given the qualitative and exploratory nature of our case study, the focus was primarily on understanding the experiences, perceptions, and attitudes of chronically ill patients during their patient journey. The semi-structured interview approach is relevant for exploratory research, as it is an effective way to gather rich data and it allows for the creation of new insights into the case under study [23]. (Page 13).

The semi-structured approach was used to obtain insights in the touchpoints in the patient journey during different periods. (Page 13 and 14).

Q12. Page 13: Line 280: It may be beneficial to split the question "What is your age and gender?" into two separate questions to improve the clarity of the data collected.

R. We have now separated the two questions to improve the clarity of the data collected.

C. 1. What is your age? and 2. What is your gender? (Page 14).

Q13. Page 16: Line 289: Add references here of that literature here.

R. We have now added the references that we used to create our coding framework.

C. We employed this deductive, thematic analysis approach since we used a coding framework for analysis that was based on concepts and definitions derived from the literature [4, 6, 13, 19]. (Page 17).

Q14. Page 16: Line 296: Is this text referencing style in line with PLoS One house style?

R. We have double checked the text referencing style and can verify that the text referencing style is in accordance with the submission guidelines of PLoS ONE.

Q15. Page 20: Line 316: Consider rephrasing "have been" to "were"

R. We thank the reviewer for this suggestion and have changed the manuscript accordingly.

C. To establish validity and reliability of our data, several measures were taken [33]. (Page 20).

Q16. Page 20: Line 323: The authors mentioned that the transcripts were returned to all participants, and no comments or corrections were received. Can you kindly provide some additional information on the timeline for this process? Specifically, how long did you wait for participants to review and return the transcripts before considering it as no comments received? This information will help readers understand the study's timeline and the data verification process.

R. We have now added a sentence in which we provide additional information on the timeline for the process of member checking.

C. The transcripts were returned to all participants and they were given one week to review and return their transcript. A reminder was sent after one week. We received no comments and corrections, all participants agreed on their transcript. (Page 21).

Q17. Page 20: Line 327: Can you please clarify the decision made by the Ethics Review Board regarding the no necessity of Medical Research Involving Human Subject (WMO) approval from a Medical Ethics Committee? It would be helpful to understand the reasons behind this decision in order to ensure that ethical considerations were thoroughly evaluated and followed in the study.

R. We apologize that we have been ambiguous about the approval of the Ethics Review Board which in turn caused confusion about the approval of the Medical Ethics Committee. We have decided to skip the sentence regarding the fact that no Medical Research Involving Human Subject approval from a Medical Ethics Committee was necessary. For this type of research, this type of approval is not required in general and as such, by including this sentence, we have been superfluous. Thus, we skipped this sentence. As a result, the text is now completely explicit about the approval of the Ethics Review Board of Catharina Hospital Eindhoven. Please see the manuscript accordingly.

C. The Ethics Review Board of Catharina Hospital Eindhoven thoroughly evaluated and approved our study design (nWMO-2021.055). We informed the participants about the study and their rights as a participant in scientific research. All participants provided oral and written informed consent.

Q18. Page 20, 21: Line 334 - 341: I have reviewed the Results section and would like to recommend restructuring this section. Specifically, I believe the information about the number of patients approached, and reasons for non-participation is more appropriate in the Methods section than in the Results section. As per scientific writing conventions, the Results section should focus on presenting the study's key findings, while the Methods section provides details about the study design, recruitment process, and other pertinent information. Therefore, I would suggest revising the Results section to focus solely on the presentation of the interview results, as well as the touchpoints and digital alternatives explored in the study.

R. We have now moved the statements related to the number of patients approached, and reasons for non-participation to the Methods section. In addition, we have moved the description of study participants to the Methods section. As such, we ensure that the Results section solely focuses on the presentation of the interview results. Please see the manuscript accordingly. 

Q19. Page 20: Line 336: In the results section, please present the interview findings in past tense since the study has already been conducted and you are reporting the results.

R. We have now ensured that the interview findings are presented in past tense. Please see the manuscript accordingly.

Q20. Page 21: Line 346 and 347: It is recommended to report the results using the appropriate scientific notation, such as the use of symbols and punctuation marks. For instance, instead of using a comma to separate the mean and standard deviation, it is more appropriate to use a semi-colon to separate them, as in M=50; SD=13.09. This notation is widely used in scientific writing and can help to ensure clarity and consistency in the reporting of results.

R. We agree with the reviewer that we did not use the appropriate scientific notation to report the results. We have now ensured that we consistently apply a semi-colon to separate the mean and standard deviation. 

C. The youngest participant was 38 years old and the oldest was 78 years old (M=50.00 years; SD=13.09). On average, participants were living with their chronic illness for 18 years at the time of the interviews (M=18.00 years; SD=11.26). (Page 12).

Q21. Overall, these revisions will help improve the clarity and consistency of your manuscript, and ensure that readers can understand the study's methods, findings, and conclusions more easily.

R. We agree with the reviewer that the suggested revisions have improved the clarity and consistency of our manuscript. As a result, we hope that the revised manuscript is now easier to understand for the readers of PLoS ONE.

---

## [Decision Letter · Decision Letter 3]

4 May 2023

The never-ending patient journey of chronically ill patients: A qualitative case study on touchpoints in relation to patient-centered care

PONE-D-22-21246R3

Dear Dr. Peters,

We’re pleased to inform you that your manuscript has been judged scientifically suitable for publication and will be formally accepted for publication once it meets all outstanding technical requirements.

Kind regards,

Edward Nicol, PhD

Academic Editor

PLOS ONE

Additional Editor Comments (optional):

Reviewers' comments:

Reviewer's Responses to Questions

**Comments to the Author**

1. If the authors have adequately addressed your comments raised in a previous round of review and you feel that this manuscript is now acceptable for publication, you may indicate that here to bypass the “Comments to the Author” section, enter your conflict of interest statement in the “Confidential to Editor” section, and submit your "Accept" recommendation.

Reviewer #3: All comments have been addressed

2. Is the manuscript technically sound, and do the data support the conclusions?

Reviewer #3: Yes

3. Has the statistical analysis been performed appropriately and rigorously? 

Reviewer #3: N/A

4. Have the authors made all data underlying the findings in their manuscript fully available?

Reviewer #3: Yes

5. Is the manuscript presented in an intelligible fashion and written in standard English?

Reviewer #3: Yes

6. Review Comments to the Author

Reviewer #3: After reviewing the manuscript, I recommend its acceptance. The authors have made careful revisions, resulting in improved readability. If accepted for publication in PloS One, I believe this manuscript will make a valuable scientific contribution to the field.

7. PLOS authors have the option to publish the peer review history of their article (what does this mean?). If published, this will include your full peer review and any attached files.

Reviewer #3: **Yes: **Imran Hameed Khaliq

---

## [Editor Report · Acceptance letter]

8 May 2023

PONE-D-22-21246R3 

The never-ending patient journey of chronically ill patients: A qualitative case study on touchpoints in relation to patient-centered care 

Dear Dr. Peters:

I'm pleased to inform you that your manuscript has been deemed suitable for publication in PLOS ONE. Congratulations! Your manuscript is now with our production department. 

Kind regards, 

on behalf of

Dr. Edward Nicol 

Academic Editor

PLOS ONE